



Atmospheric
Chemistry
and Physics

# Compliance and port air quality features with respect to ship fuel switching regulation: a field observation campaign, SEISO-Bohai

**Yanni Zhang**[1,2], **Fanyuan Deng**[1,2], **Hanyang Man**[1,2], **Mingliang Fu**[1,2,3], **Zhaofeng Lv**[1,2], **Qian Xiao**[1,2], **Xinxin Jin**[1,2], **Shuai Liu**[1,2], **Kebin He**[1,2], and **Huan Liu**[1,2]

[1]State Key Joint Laboratory of ESPC, School of Environment, Tsinghua University, Beijing 100084, China
[2]State Environmental Protection Key Laboratory of Sources and Control of Air Pollution Complex, Beijing, 100084, China
[3]State Key Laboratory of Environmental Criteria and Risk Assessment (SKLECRA), Chinese Research Academy of Environmental Sciences, Beijing, 100012, China

**Correspondence:** Huan Liu (liu_env@tsinghua.edu.cn)

**Abstract.** Since 1 January 2017, ships berthed at the core ports of three designated "domestic emission control areas" (DECAs) in China should be using fuel with a sulfur content less than or equal to 0.5 %. In order to evaluate the impacts of fuel switching, a measurement campaign (SEISO-Bohai) was conducted from 28 December 2016 to 15 January 2017 at Jingtang Harbor, an area within the seventh busiest port in the world. This campaign included meteorological monitoring, pollutant monitoring, aerosol sampling and fuel sampling. During the campaign, 16 ship plumes were captured by the on-shore measurement site, and 4 plumes indicated the usage of high-$S_F$ ($S_F$ refers to the sulfur content of marine fuels). The average reduction of the mean $\Delta NO_x/\Delta SO_2$ ratio from high-sulfur plumes (3.26) before 1 January to low-sulfur plumes (12.97) after 1 January shows a direct $SO_2$ emission reduction of 75 %, consistent with the sulfur content reduction (79 %). The average concentrations of $PM_{2.5}$ (particulate matter with a diameter less than 2.5 μm), $NO_x$, $SO_2$, $O_3$ and CO during campaign were 147.85 μg m$^{-3}$, 146.93, 21.91, 29.68 ppb and 2.21 ppm, respectively, among which $NO_x$ reached a maximum hourly concentration of 692.6 ppb, and $SO_2$ reached a maximum hourly concentration of 165.5 ppb. The mean concentrations of carbonaceous and dominant ionic species in particles were 6.52 (EC – elemental carbon), 23.10 (OC – organic carbon), 22.04 ($SO_4^{2-}$), 25.95 ($NO_3^-$) and 13.55 ($NH_4^+$) μg m$^{-3}$. Although the carbonaceous species in particles were not significantly affected by fuel switching, the gas and particle pollutants in the ambient air exhibited clear and effective improvements due to the

implementation of low-sulfur fuel. Comparison with the prevailing atmospheric conditions and a wind map of $SO_2$ variation concluded a prompt $SO_2$ reduction of 70 % in ambient air after fuel switching. Given the high humidity at the study site, this $SO_2$ reduction will abate the concentration of secondary aerosols and improve the acidity of particulate matter. Based on the enrichment factors of elements in $PM_{2.5}$, vanadium was identified as a marker of residual fuel ship emissions, decreasing significantly by 97.1 % from 309.9 ng m$^{-3}$ before fuel switching to 9.1 ng m$^{-3}$ after regulation, which indicated a crucial improvement due to the implementation of low-sulfur fuels. Ship emissions were proven to be significantly influential both directly and indirectly on the port environment and the coastal areas around Bohai Bay, where the population density reaches over 650 people per square kilometer. The results from this study report the positive impact of fuel switching on the air quality in the study region and indicate a new method for identifying the ship fuel type used by vessels in the area.

## 1 Introduction

Maritime transport is an important source of pollutants globally; thus, it is one of the well-established culprits regarding the adverse effects of ship emissions on air quality (Eyring et al., 2005, 2010; Endresen et al., 2003; Fridell et al., 2008; Jalkanen et al., 2009; Liu et al., 2016; Viana et al., 2014), climate (Lauer et al., 2007; Tronstad Lund et al., 2012; Liu

et al., 2016) and human health (Campling et al., 2013; Corbett et al., 2007; Winebrake et al., 2009). Estimations show that ships contribute 15 % of the global $NO_x$ emissions in addition to 4 %–9 % of the global $SO_2$ emissions (Eyring et al., 2010). In the EU 27, ships emitted 2.8 million tons of $NO_x$, 1.7 million tons of $SO_2$ and 0.2 million tons of $PM_{2.5}$ in 2005, from which approximately 70 % was emitted within 200 nmi of the coast of EU member states (Campling et al., 2013). From 2006 to 2009, $NO_x$ emissions from ships rose by approximately 7 % in the Baltic Sea, while $SO_2$ and $PM_{2.5}$ emissions decreased by 14 % and 20 %, respectively, which was mainly caused by fuel regulations that came into effect in 2006 (Jalkanen et al., 2014). In 2011, ship emissions in Europe were estimated to be responsible for 2.0 million tons of $NO_x$, 1.2 million tons of $SO_2$ and 0.2 million tons of $PM_{2.5}$ (Jalkanen et al., 2016). According to the United Nations Conference on Trade And Development (UNCTAD, 2017), the volume of the world's seaborne trade grew by 66 % between 2000 and 2015. As global commerce expands, ocean-going vessels consume more fuels – generally low-quality residual fuels containing high concentrations of sulfur and heavy metals (Lack et al., 2011) – which differ greatly from inland fuel usage. In China, the average sulfur content of marine fuel (average $S_F$) was 2.43 % (by mass, i.e., 24 300 ppm) before regulation (Liu et al., 2016), which was much higher than the sulfur content restriction of 10 ppm that was applied to inland fuels (Chinese national standards GB 19147-2013 and GB 17930-2013). This makes ships one of the prominent contributors of pollutant emissions in major port cities (Lai et al., 2013; HKEPD, 2014; Zhao et al., 2013). Estimations of ship emissions within 200 nmi of the Chinese coast have shown that ships accounted for an annual increase of up to 5.2 $\mu g\,m^{-3}$ $PM_{2.5}$ in eastern China, which influenced the air quality not only in coastal areas but also in inland areas that are hundreds of kilometers from the sea (Lv et al., 2018). In 2010, ships contributed 12.0 % of the total $SO_x$ emissions, 9.0 % of the total $NO_x$ emissions and 5.3 % of the total $PM_{2.5}$ emissions in Shanghai (Fu et al., 2012). Furthermore, 14.1 % of $SO_2$ emissions, 11.6 % of $NO_x$ emissions and 3.6 % of $PM_{2.5}$ emissions within the Pearl River Delta region, China, were attributed to ships in 2013 (Li et al., 2016).

These situations have drawn a lot of attention regarding coastal air pollution and related emission control strategies, such as scrubbers. However, recent research has also reported the potential pollution of surface waters by ship emissions due to certain methods of treating ship exhausts (Hassellöv et al., 2013; Stips et al., 2016; Turner et al., 2017, 2018), indicating that the exhaust aftertreatment may not be the best choice of ship emission reduction. The International Maritime Organization (IMO), the European Union and the US have implemented regulations in an effort to reduce ship emissions, among which fuel quality regulation has proven potent in many countries for addressing the issue of sulfur oxides ($SO_x$) and particulate matter (PM) emissions. The IMO has regulated the $S_F$ on a global scale from the current value of 3.5 % to 0.5 % by 2020, and has implemented more stringent legislation in designated emission control areas (ECAs), which with respect to $SO_x$ emissions include the Baltic Sea, the North Sea, the English Channel, and coastal waters around the Canada, US and the US Caribbean Sea. The $S_F$ allowed in ECAs was 1 % in 2010 and has been reduced to 0.1 % since 1 January 2015 (IMO, 2008). Estimation in European seas shows that this IMO limitation of 0.1 % $S_F$ in ECAs would reduce $SO_2$ emissions by 82 % by 2020 and decrease the amount of $SO_2$ by a further 23 000 tons by 2030 (Campling et al., 2013). This assumption is supported by other comparable results from subregion assessment and in site measurements (Matthias et al., 2010; Viana et al., 2015; Zetterdahl et al., 2016). Likewise, EU Directive 2012/33/EU has demanded that ships at berth in European Union ports use fuels with a $S_F < 0.1$ % since December 2012, which reduced $PM_{2.5}$ emissions from ships by about 50 % between 2007 and 2012 in Venice, Italy (Contini et al., 2015). Beginning in July 2009, the US state of California introduced legislations limiting vessels operating within 24 nautical miles (44 km) of the California coastline to the use of marine gas oil (MGO) or marine diesel oil (MDO) with a maximum $S_F$ of 1.5 % or 0.5 %, respectively (from January 2012 $S_F$ content was restricted to < 0.1 %) (CARB, 2009). As a result, a clear improvement in the air quality was observed at the Port of Oakland and in the surrounding San Francisco Bay area in 2010 (Tao et al., 2013). Lack et al. (2011) also reported that fuel quality regulation along with speed restriction in California could potentially generate an 88 % reduction in gaseous and particle pollutant concentrations.

Based on the abovementioned widely acclaimed fuel quality regulations, China promulgated the implementation of the ship emission control area in the Pearl River Delta, the Yangtze River Delta and the Bohai Rim (Beijing–Tianjin–Hebei area) (MOT, 2015) in 2015, designing three DECAs with phased $S_F$ requirements. Since 1 January 2017, ships berthed at the core port of these three DECAs should be using fuel with a $S_F$ less than or equal to 0.5 %. This new regulation provides the opportunity to measure the influence of limiting the $S_F$ on the magnitude of ship emissions in China. However, compared with fuel regulations in Canada, Europe and the US, which have undergone years of enforcement and optimization, the Chinese regulation is incipient in clauses and vague in terms of supervision. The possible effects of ship emission control are indeed compelling, but they are also difficult to evaluate due to the variability of complicated local emission sources and the complexity of fleet management. Until now, much of the previous research on the subject of ship emission control has been restricted to limited comparisons of emissions, which have failed to specify the compliance of vessels or a practical method to indicate it.

In order to explore the methods used to capture fuel-related emission change and the impact on the air quality due to fuel switching, we selected the Bohai Rim (Beijing–

Tianjin–Hebei area) as the study site and conducted in situ measurements of meteorological parameters and pollutants along with chemical analyses of sampled fuels and aerosols, which are all typical methods utilized within the field of air quality measurement. The campaign ran from 27 December 2016 to 15 January 2017, covering the primary implementation period of the new regulation. By comparing ship emissions and air quality before and after fuel switching, this paper sheds light on the potential emission reduction effects of the enforced regulation. Meanwhile, certain features in ship plumes were found to be related to fuel type, providing another angle for supervising ship fuels in practice. This may be helpful in the actual implementation and management of the new regulation.

## 2 Methodology

### 2.1 Field measurement

#### 2.1.1 Measurement site

The measurement station ($39.204576°$ N, $119.004028°$ E) for the "Shipping Emission and Impacts by Switching Oil in Bohai Bay" (SEISO-Bohai) campaign is located on the corner of a main navigational channel to the third pool in Jingtang Harbor (hereafter referred to as JT), on the property (and with the support of) the Hebei Tangshan Harbor Economic Development Zone Office (Fig. 1). Located in Bohai Bay, JT belongs to the Port of Tangshan, one of the core ports in the designated DECAs. China Port Yearbook (2017) reported a total throughput of 520 million tons in the Port of Tangshan, from which JT handled over 270 million tons. The population density around JT and the surrounding "Port Economic Development Area" is high, with over 650 people per square kilometer. JT is described in more detail elsewhere (Xiao et al., 2018). The station consists of a measurement container, which has a small meteorological monitoring station placed on the roof, and aerosol sampling instruments.

#### 2.1.2 Meteorological monitoring

A small meteorological monitoring station was placed on the roof of the container and obtained temperature (°C), relative humidity (%), wind speed (m s$^{-1}$), wind direction and radiation intensity every 1 min, from 28 December 2016 to 15 January 2017. Abrupt high-temperature values were subtracted from the results as they were obviously invalid data, for example, when an instrument indicated $40 °C$ for ambient temperature in winter.

#### 2.1.3 Particle and gas monitoring

Continuous concentrations of six gases (NO, $NO_2$, $NO_x$, $SO_2$, and $O_3$ in ppb, and CO in ppm) were measured every 1 min, and $PM_{2.5}$ and $PM_{10}$ (in µg m$^{-3}$) were measured every 1 h, using a Sailhero XHAQMS3000 air quality continuous monitoring system, from 28 December 2016 to 13 January 2017. Monitoring modules consist of NO, $NO_2$ and $NO_x$ measurement using an analyzer, $SO_2$ detection using a UV fluorometer, CO using IR absorption, $O_3$ using UV spectrophotometry and particles using $\beta$-ray absorption (ISO 10473:2000). The instrument displayed a short, erroneous measurement at the beginning of the campaign, possibly due to unskilled operation, which resulted in some negative values for gas pollutants and overexaggerated values for $SO_2$, CO and $O_3$. The instrument was immediately repaired, and the abovementioned values were removed to ensure the accuracy of the data. Invalid values of $O_3$ occurred sporadically during the campaign, appearing as a sinusoid fluctuation below 10 ppb, and were subtracted from the results. It should be mentioned that the air quality of Xinli Primary School (hereafter referred to as XL, an official air quality monitor site as shown in Fig. 1) was provided by an official air quality monitor and was used as a control (see in http://www.aqistudy.cn, last access: 15 April 2017).

#### 2.1.4 Particle samples

The campaign resulted in the collection of 14 valid particle samples: 2 parallel samples were collected per day before 31 December 2016 and 1 sample was collected per day after that. The filters were exposed for 23 h (normally from 16:30 to 15:30 LST, local standard time, the next day and were labeled according to the end date) on 80 mm (diameter) preferred quartz microfiber filters (CHM QF1 grade) using a Laoying Model 2030 TSP sampler. All samples were immediately put into their original polyethylene plastic boxes, wrapped in two layers of prebacked tinfoil, and then subsequently housed in a refrigerator. In order to avoid any possible contamination of the samples, all of the abovementioned procedures were strictly quality-controlled.

### 2.2 Chemical analysis

#### 2.2.1 Carbon analysis

A $0.55 cm^2$ section of each exposed filter and blank filters were measured for organic carbon (OC) and elemental carbon (EC) concentrations using the thermal optical transmission method in a DRI 2001 organic carbon/elemental carbon (OC/EC) analyzer. OC and EC values were determined via the Interagency Monitoring of Protected Visual Environments protocol (referred to as the IMPROVE A method). Samples were heated in a completely oxygen-free helium atmosphere, using four increasing temperature steps to remove all of the OC on the filter, during which part of the OC was pyrolyzed. Then the pure helium eluent was switched into a 10 % oxygen/helium mixture in the oven and stepped to $800 °C$ for EC determination. OC and EC were detected using a flame ionization detector after oxidation to carbon diox-

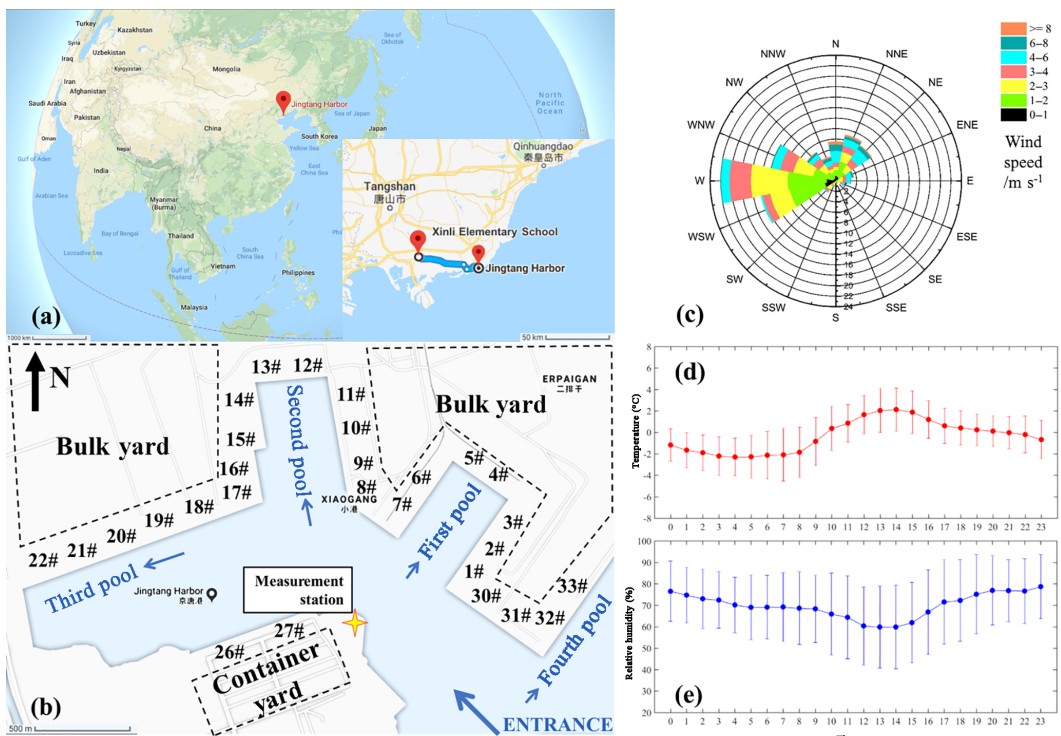

**Figure 1. (a)** The location of Jingtang Harbor (JT), and the location of an official air quality monitoring station, Xinli Elementary School (XL; map inset). **(b)** The location of the measurement station (yellow marker) and the distribution of pools, berths and loading areas in the port domain. Wind rose **(c)**, daily variation of temperature **(d)** and relative humidity **(e)** from the measurement station during the period from 28 December 2016 to 13 January 2017.

ide and then reduced to methane. The detection limit of this analysis was $0.82 \, \mu g \, cm^{-2}$ for OC and $0.2 \, \mu g \, cm^{-2}$ for EC.

### 2.2.2 Ion analysis

A $50 \, cm^2$ section of each exposed filter and blank filters were extracted using $10 \, mL$ of ultra-pure water in an ultrasonic bath at $4 \, °C$ for $30 \, min$. Inorganic ions of $Na^+$, $NH_4^+$, $K^+$, $Mg^{2+}$, $Ca^{2+}$, $Cl^-$, $NO_3^-$ and $SO_4^{2-}$ were analyzed using a DIONEX ICS-2100 ion chromatograph. The ion chromatograph system was calibrated using a standard solution before the samples were run. Data obtained from a sample were compared to data from the known standard, achieving the identification and quantification of sample ions. The detection limit was $0.1 \, \mu g \, L^{-1}$.

### 2.2.3 Element analysis

A $20 \, cm^2$ section of each exposed filter and blank filters were digested using $25 \, mL$ of an $8 \%$ HCl/$3 \%$–HNO$_3$ solution in an ultrasonic bath at $69 \, °C$ for $3 \, h$. The solutions were cooled, vortex mixed and then placed in a centrifuge at $2800 \, rpm$ for $15 \, min$ to settle any insoluble particle, from which a $5 \, mL$ aliquot was taken for the analysis of the following 30 elements using an X Series 2 ICP-MS mass spectrometer: Be,

Na, Mg, Al, K, Ca, Ti, V, Cr, Mn, Fe, Co, Ni, Cu, Zn, As, Se, Sr, Mo, Ag, Sn, Ba, La, Ce, Hg, TI, Pb, Th and U. Measured Be concentrations were generally $0 \, \mu g \, m^{-3}$ throughout the sampling period, and this species was consequently removed from the results. Cr was also removed, as the blank value exceeded most of the sample results. Several concentrations of Cd and Mo were below detection and were also removed.

### 2.3 Ship plume events

Identifying a "ship plume event" using direct and simultaneous measurements of trace gases with in situ instruments aims at the surveillance of emissions and fuel types utilized by passing ships. As the measurement site is in the vicinity of the channel and the berths, when wind directions are favorable for measurements ship plumes passing the instrument cause a distinctive change in the measured components against background concentrations; these changes are defined as a ship plume event. Several studies have confirmed that synchronic variation in pollutant concentrations can be used to identify the occurrence of ship plume events from observation made near the harbor (Alföldy et al., 2013; Ault et al., 2010; Contini et al., 2015; Gao et al., 2016; Lu et al., 2006; Kattner et al., 2015): $SO_2$, $NO_x$, $CO_2$, BC, $PM_{2.5}$ concentrations and number concentrations of aerosol parti-

**Table 1.** Observations of trace gases (ppb) and molar $\Delta NO_x/\Delta SO_2$ ratios ($ppb\,ppb^{-1}$) in ship plumes.

| # | Date and time | Wind direction | Wind speed $(m\,s^{-1})$ | Max $NO_x$ | Max $SO_2$ | Regression $\Delta NO_x/\Delta SO_2$ | Background* $NO_x/SO_2$ | $N$ | $R^2$ |
|---|---|---|---|---|---|---|---|---|---|
| 1 | 28 Dec 23:20–23:44 | Northwest | 5.26 | 226 | 127.6 | $1.92 \pm 0.38$ | 3.16 | 6 | 0.83 |
| 2 | 29 Dec 04:08–04:32 | West | 2.1 | 239.7 | 134.7 | $0.92 \pm 0.27$ | 2.06 | 6 | 0.68 |
| 3 | 29 Dec 06:52–07:40 | West | 2.3 | 277.1 | 106.5 | $1.02 \pm 0.29$ | 2.06 | 12 | 0.51 |
| 4 | 30 Dec 08:36–09:12 | West–northwest | 1.5 | 161.5 | 35.4 | $1.95 \pm 0.4$ | 3.03 | 9 | 0.74 |
| 5 | 30 Dec 09:16–09:52 | West–northwest | 1.8 | 306.6 | 60.3 | $3.79 \pm 1.21$ | 3.03 | 9 | 0.52 |
| 6 | 31 Dec 07:48–08:12 | West | 1.2 | 331.1 | 42 | $17.89 \pm 2.25$ | 4.99 | 6 | 0.93 |
| 7 | 31 Dec 21:40–22:20 | West | 1.3 | 551.8 | 50 | $10.14 \pm 1.55$ | 4.99 | 10 | 0.82 |
| 8 | 31 Dec 22:28–23:32 | West | 1 | 438.3 | 29.1 | $14.48 \pm 2.43$ | 4.99 | 16 | 0.7 |
| 9 | 5 Jan 15:36–16:08 | East–northeast | 4.1 | 242.1 | 72.6 | $3.47 \pm 0.26$ | 1.29 | 8 | 0.96 |
| 10 | 5 Jan 18:24–18:56 | East–northeast | 2.2 | 122.9 | 72.6 | $5.81 \pm 1.25$ | 1.29 | 8 | 0.75 |
| 11 | 8 Jan 00:00–00:28 | North–northeast | 2.5 | 176.8 | 17.6 | $17.45 \pm 4.56$ | 3.25 | 8 | 0.66 |
| 12 | 9 Jan 04:16–04:48 | North | 3.7 | 183.7 | 28.3 | $6.43 \pm 1.32$ | 1.67 | 8 | 0.76 |
| 13 | 9 Jan 23:12–23:56 | West | 2.8 | 226 | 18.9 | $10.88 \pm 1.43$ | 1.67 | 11 | 0.85 |
| 14 | 10 Jan 05:08–05:32 | North | 6.1 | 158.4 | 47.9 | $4.01 \pm 1.08$ | 2.13 | 6 | 0.72 |
| 15 | 11 Jan 18:16–18:48 | West–southwest | 1.6 | 115.3 | 18 | $12.00 \pm 1.31$ | 2.11 | 8 | 0.92 |
| 16 | 13 Jan 14:20–14:42 | Northwest | 6.4 | 204.9 | 32.8 | $7.95 \pm 2.13$ | 2.83 | 6 | 0.72 |

* Concentrations provided by an air monitoring station at Xinli Primary School (XL).

cles increase simultaneously at the onset of the ship plume, whereas the $O_3$ concentration decreases due to its reduction reaction with NO to form $NO_2$.

Nitrogen compounds were abundant in the atmosphere in JT due to the heavy traffic, whereas the source of high $SO_2$ emissions was rather simple. Therefore, $SO_2$ peaks, or $SO_2$ episodes, were used as an indicator of recent anthropogenic emissions. The background $SO_2$ per day was set as the daily lowest concentration, and any enhancement that was higher than 3 ppb was marked as the time stamp of a possible ship emission event. For these time stamps, peaks in $NO_x$ along with simultaneous valleys in $O_3$ were then identified in valid data. The signals were only validated when there were significant peaks and clearly determinable backgrounds. Finally a ship plume event was marked if the existence of ships was positive in the upwind direction of those signals. The combination of the trace gas peak time, the wind direction and the ship traffic information (times when ships left the harbor or berthed) provided by marine administration in the port enabled the identification of the ship responsible for the plume. For example, a ship plume event was identified on 5 January 2017 from 15:36 to 16:08 (Fig. 2). The times and conditions associated with 16 positively identified ship plume events are listed in Table 1. Several situations made it difficult to identify a ship plume event in our measurement. Firstly, there was a period when not many ships entered the port due to the New Year holiday and poor visibility from 1 to 4 January 2017. Secondly, the prevailing wind direction indicated that our plumes would mainly have been from the second and the third pools, where approximately half of the berths were actually under construction and not in use. These conditions meant fewer plumes than expected. Furthermore, rapid changes in the wind direction was sometimes unfavorable with respect to the instruments' ability to capture the ship plumes. Thirdly, the port is generally quite polluted (on over 50 % of days, the $PM_{2.5}$ concentration was above $115\,\mu g\,m^{-3}$, see Sect. 3.1.1), and the rather high concentration of existing pollutants may mask the existence of a ship plume event. Moreover, the measurement site is also in the vicinity of high land-based vehicular emissions (truck traffic), which may be another interference.

The $SO_2$ to $CO_2$ ratio in ship plumes is widely used as an indicator of $S_F$ (Yang et al., 2016; Kattner et al., 2015; Balzani Löv et al., 2014). However, in this study we aimed to explore another applicable indicator for $S_F$ in China, as the concentration of $CO_2$ is often excluded from ambient air measurements due to the fact that the ambient air quality standards (China national standard GB 3095-2012) stipulates which six pollutants should be monitored and excludes $CO_2$. The ratio of ship-emitted $NO_x$ to $SO_2$, i.e., the enhancement of $NO_x$ to $SO_2$ in observations ($\Delta NO_x/\Delta SO_2$), is correlated with the fuel type, and rises if ships switch to low-sulfur fuel (McLaren et al., 2012; Sinha et al., 2003). Moreover, based on the ship information provided by JT, the size of the berths and the design of the port, we found that ships in JT, especially those related to identified plumes, were mainly consistent with respect to size, which implies similar $NO_x$ emissions in these plumes (IMO, 2015). Therefore, the $\Delta NO_x/\Delta SO_2$ ratio is appropriate to indicate the $S_F$ of ships in JT. Data from ship plume events were averaged every 4 min, and a suitable baseline point was set as the background concentration; this background was taken either before or after the ship plume event in question. Then the $\Delta NO_x/\Delta SO_2$ ratios were calculated via a two-sided linear least-squares re-

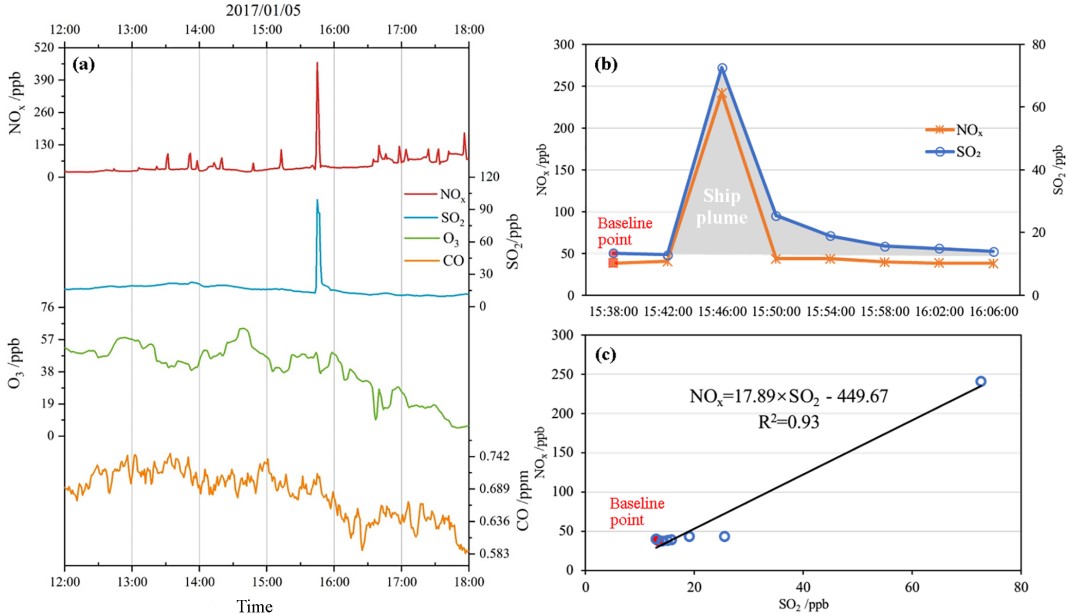

**Figure 2.** Marine vessel plume number 9 showing the **(b)** ship plume interval identified from **(a)** NO, $SO_2$, $O_3$ and CO concentrations measured in Jingtang Harbor (JT) from 12:00 to 18:00 LST, 5 January 2017, and **(c)** the linear regression method for the determination of the $NO_x/SO_2$ ratio.

gression of $NO_2$ to $SO_2$ using all points within each plume event, including the baseline point (Fig. 2). This method is similar to that used for the determination of emission ratios (McLaren et al., 2012) or emission factors (Williams et al., 2009) in ship plumes.

## 2.4 Properties of fuel samples

Intermediate fuel oil (IFO), also known as heavy fuel oil, is typically used by marine vessels. IFO is the petroleum product left after all of the other fractions from crude oil have been distilled. This product has a high density, carbon/hydrogen ratio and sulfur content (varying from 2 % to 5 %) compared with gas and oil products used by other means of transportation. In addition, IFO contains high concentrations of organics and metals from the original crude oil. IFOs are categorized into IFO380, IFO180 and IFO60 by their maximum viscosity measured at 50 °C, and the fuel quality is generally better as the viscosity decreases (Table 2). Recent onboard and in situ measurements have revealed that high-$S_F$ fuel generally creates higher sulfur, particle and soot emissions (Celo et al., 2015; Contini et al., 2015; Cooper, 2003; Fridell et al., 2008; Lack et al., 2011; Moldanová et al., 2009; Petzold et al., 2010; Sinha et al., 2003; Winnes and Fridell, 2010; Winnes et al., 2016). A significant metal contribution from residual fuel combustion has also been noted (Lake et al., 2004), in addition to the contribution to the emission (Kweon et al., 2003; Lack et al., 2011; Lee et al., 1998) and formation (Ault et al., 2010) of particulate organic matter.

Research has shown that ships in sulfur emission control areas (SECAs) switch to marine diesel fuel (MDO), a cleaner fuel typically used to meet the requirement of many fuel quality regulations and emission limits. Compared with IFOs, MDOs have a lower density, a lower carbon/hydrogen ratio, and a lower nitrogen ($\sim 10$ % of IFOs), sulfur ($\sim 30$ % of IFOs) and heavy metal (significant reduction) content (Table 2). Due to their low $S_F$, these cleaner fuels have proven to be better with respect to emissions, and promise an overall reduction of emissions (Cooper and Gustafsson, 2004) and the further issues of acidification and eutrophication (Bengtsson et al., 2011; Fridell et al., 2008; Sinha et al., 2003). Another record worth mentioning is that hybrid fuels, which blend IFO and other low-$S_F$ fuels to comply with $S_F$ regulations, are widely used by ships operating in SECAs (Winnes et al., 2016; Zetterdahl et al., 2016), as the price of distillate fuels is an obstacle with respect to ships completely abandoning IFOs. However, ISO 8217:2017, the current benchmark regarding the quality of marine fuels on the market, does not specify any limits on the physical and chemical parameters of hybrid fuels. This causes large uncertainty with respect to the quality of these fuels, as there are no formal standards for the quality of hybrid fuels except the restrictions on $S_F$. As Table 2 shows that the content of metals in hybrid fuels are between those of IFOs and MDOs due to the blending; however, the density, carbon, hydrogen and nitrogen are consistent with those of IFOs, indicating a quality similar to that of IFOs, which has been proven to be less of an improvement on particle emission and health impacts than totally abandoning IFOs (Winnes et al., 2016).

**Table 2.** Components of intermediate fuel oil (IFO), hybrid fuels, and marine gas oils (MGOs) and marine diesel oils (MDOs).

| Fuel for main engine | | IFOs | | | Hybrid fuels | | | | MGOs & MDOs | | | | |
|---|---|---|---|---|---|---|---|---|---|---|---|---|---|
| | | Celo et al. (2015) | | | Winnes et al. (2016) | | Zetterdahl et al. (2016) | | Celo et al. (2016) | Winnes et al. (2016) | This study | | |
| | | IFO380 | IFO180 | IFO60 | | | | | | | Ship A | Ship B | Ship C |
| Density at $15\,°C$ ($kg\,m^{-3}$) | | 988 | 970.7 | 957.6 | 988.7 | 943.3 | 982.5 | 892.8 | 854.3 | 846.4 | 848.2 | 853.1 | 846.3 |
| w % | S | 2.7 | 2.23 | 1.22 | 0.96 | 0.58 | 0.48 | 0.092 | 0.119 | 0.1 | 0.38 | 0.08 | 0.065 |
| | C | 86.26 | 85.71 | 87.22 | 87.93 | 87.13 | 88.4 | 87.4 | 86.85 | 86.29 | 85.16 | 84.78 | 86.83 |
| | H | 11.26 | 10.51 | 11.05 | 10.68 | 12.11 | 10.9 | 12.8 | 12.97 | 13.54 | 13.07 | 13.21 | 13.15 |
| | N | 0.39 | 0.41 | 0.38 | 0.42 | 0.3 | 0.52 | 0.044 | 0.026 | <0.1 | 0.027 | 0.026 | 0.01 |
| $mg\,kg^{-1}$ | Na | 22.66 | 15.74 | – | 11 | 7 | – | – | – | <1 | 1.3 | <1 | <1 |
| | Al | 7.06 | BD | – | 20 | 16 | – | – | – | <1 | BD | BD | BD |
| | Ti | 2.36 | 3.12 | – | – | – | – | – | – | <1 | BD | BD | BD |
| | V | 133.8 | 109.4 | 38 | 20 | 6 | 5 | 1 | BD | <1 | BD | BD | BD |
| | Fe | 31.44 | 20.35 | – | 9 | 7 | 1 | 2 | – | <1 | 2.7 | <1 | <1 |
| | Ni | 63.2 | 50.3 | 21 | 15 | 9 | 33 | 2 | BD | <1 | BD | BD | BD |
| | Cu | 29.51 | BD | – | – | – | – | – | – | <1 | BD | BD | BD |
| | Zn | 2.1 | BD | 2.2 | 1 | 2 | – | <1 | 2.7 | <1 | BD | BD | BD |

BD refers to "below the detection limit". "–" refers to "not reported"

In order to establish the fuel type ships were using after the implementation of fuel restrictions in JT, three fuel samples were taken from three respective vessels berthed in JT on 14 January 2017, and the fuel properties and chemical composition of the fuels were analyzed according to the petroleum industry standard (SH) and the national standard (GB) of China.

## 2.5 Backward trajectory analysis

Back trajectories were used to identify the origin and potential influences of different source regions on the vanadium (V) concentrations during each sampling day. The 24 h back trajectories of the air mass during each sampling day were computed at 500 m a.g.l. (above ground level) using the HYSPLIT 4 model (NOAA, 2013). The Global Data Assimilation System (GDAS) meteorological data were used as input. Trajectories began at 08:00 UTC (16:00 LST, consistent with the sampling period) and were calculated every 6 h.

## 2.6 Other parameters

The enrichment factor (EF) was used for the general evaluation of influences of anthropogenic sources on the elemental contents of particles (Zhao et al., 2013) and is calculated following Eq. (1):

$$EF = (X/R)_{\mathrm{aerosol}}/(X/R)_{\mathrm{crust}}, \tag{1}$$

where $(X/R)_{\mathrm{aerosol}}$ is the concentration ratio of the element of interest, $X$, to the reference element, $R$, in aerosol, and $(X/R)_{\mathrm{crust}}$ is the concentration ratio of $X$ to $R$ in crust. We used the composition of the continental crust from Wedepohl (1995) and used Al as the reference element $R$. Species with EF values less than 10 usually indicate a major crustal source, whereas species with high EF values probably indicate a significant anthropogenic source.

The sulfur oxidation ratio (SOR) and the nitrogen oxidation ratio (NOR) are used to elucidate the $SO_4^{2-}$ and $NO_3^-$ contribution (Ohta and Okita, 1990; Ostro, 1995; Wang et al., 2005) according to the following equations:

$$SOR = [SO_4^{2-}]/\left([SO_4^{2-}] + [SO_2]\right) \tag{2}$$

$$NOR = [NO_3^-]/\left([NO_3^-] + [NO_2]\right), \tag{3}$$

where the square brackets represent molar concentrations in units of $mol\,m^{-3}$. A SOR/NOR value above 0.1 indicates a photochemical redox reaction of $SO_2$ or $NO_x$ in ambient air. Higher SOR and NOR values indicate larger amounts of secondary sulfate and nitrate formation (Khoder, 2002).

## 3 Results

### 3.1 Impacts on port air quality from fuel switching

#### 3.1.1 $SO_2$ reduction in the polluted port area

The climate of JT is strongly influenced by the sea breeze. The mean relative humidity during campaign was 69.4 % (ranging from 21.8 % to 99.9 %), while the mean temperature was $-0.6\,°C$. Temperature exhibited a clear diurnal cycle: it was lowest before dawn ($-2.3\,°C$), rose after sunrise (07:00 LST) and reached a maximum (14 °C) at

14:00 LST (Fig. 1). The prevailing wind directions were west (23.4 %) and north-northwest (13.0 %), and the wind speed mainly ranged between 1 and 4 m s$^{-1}$ (2.7 m s$^{-1}$ on average). Coastal meteorological patterns, such as those mentioned above, play an important role in the dispersion, transformation, accumulation or removal of air pollutants (Gariazzo et al., 2007).

During the campaign, the day-to-day variation in emission was large due to variation in both the complicated sources and removal in JT; however, the data generally exhibited a heavily polluted environment in JT. As the primary pollutant at site, PM$_{2.5}$ concentrations were used to classify the local pollution level. On over 50 % of days, the PM$_{2.5}$ concentration was above 115 µg m$^{-3}$, which is the Grade IV criterion of China's daily air quality standard (HJ 633-2012) (Fig. 3), and the mean concentration during the campaign (147.85 µg m$^{-3}$) was much higher than that of city area of Tangshan (117.9 µg m$^{-3}$) during the same season (Zhang et al., 2017). The PM$_{2.5}$ concentration was even three times the wintertime PM$_{2.5}$ concentration observed in Hong Kong (Gao et al., 2016), and twice that reported at the Yangshan Port, Shanghai (Zhao et al., 2013). This suggests severe air pollution in JT, which is understandable as winter (December–February) is the most polluted period in Tangshan due to the higher emissions and unfavorable atmospheric conditions (e.g., lower mixing heights and more frequent temperature inversions). Since December 2013, when an official air quality monitor station began to operate in the area (http://www.aqistudy.cn, last access: 15 April 2017), the reported PM$_{2.5}$ concentration during the cold season has always exceeded 100 µg m$^{-3}$. This situation indicates the necessity for implementing appropriate particle emission reduction measures. Gas pollutants have also been found to be abundant in JT due to the heavy traffic. Average concentrations of NO$_x$, SO$_2$, O$_3$ and CO were 146.93, 21.91, 29.68 ppb and 2.21 ppm, respectively. NO$_x$ seemed to be an issue in the port, showing a high maximum hourly concentration of 692.6 ppb during the campaign, whereas SO$_2$ reached a maximum of 165.5 ppb. Peak levels of NO$_x$ and SO$_2$ were mainly linked with ship activities, as the measurement site was very close to the channel and the berths. A lower level of O$_3$ was observed in JT compared with Yangshan Port in Shanghai (Zhao et al., 2013), and a clear diurnal cycle of O$_3$ was noted: the concentration rise at daytime (29.18 ppb) and fell at night (16.38 ppb). The combined influence of coastal meteorology was responsible for this cycle to some degree. During daytime, photochemical reactions and transportation of ozone-rich air increases O$_3$, whereas the reaction with NO and dry deposition destroy O$_3$ at night. Research has shown that O$_3$ can be totally destroyed if the NO source is large enough (Finlayson-Pitts and Pitts, 2000), and as our study site was located in a busy port, our data verify this finding (e.g., the O$_3$ concentration was approximately 0 ppb at 21:00 LST on 4 January 2017). Peaks of CO and NO$_x$ coexisted to some

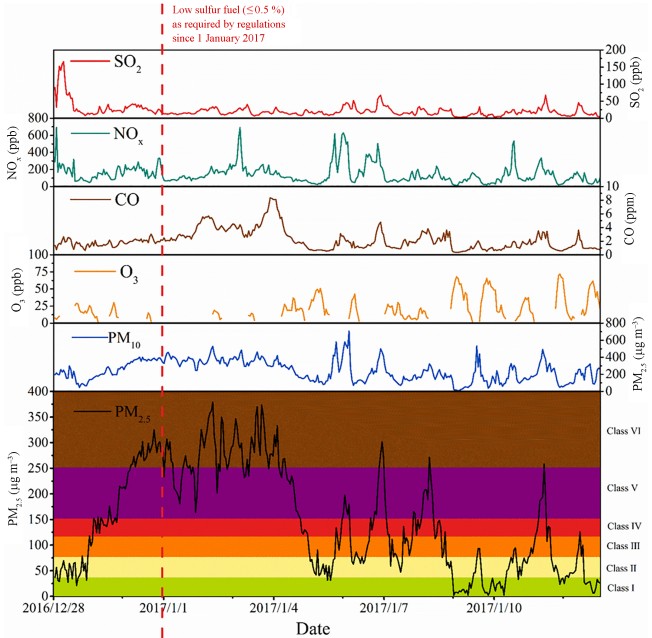

**Figure 3.** Hourly SO$_2$, NO$_x$, CO, O$_3$, PM$_{10}$ and PM$_{2.5}$ concentrations measured in Jingtang Harbor (JT) from 28 December 2016 to 13 January 2017.

degree, but overall there was no evident pattern for CO due to complex local combustion emissions.

The SO$_2$ level reduced notably with the maximum hourly concentration dropping from 165.5 ppb before 1 January 2017 to 67.4 ppb after this date, with similar vessel activity. The variation in Fig. 4 confirms this distinct reduction in JT. SO$_2$ exhibited a prompt drop from 77 to 20 ppb on 30 December 2016 compared with the high and steady concentration at XL (the control group from upwind of JT). This SO$_2$ reduction was mainly attributed to local sources, as JT was under the influence of the prevailing atmospheric conditions from XL via diffusion and transmission (NO$_x$ and PM$_{2.5}$ covaried at both sites), where SO$_2$ displayed little change. More precisely, the reduction was most likely a direct response to fuel switching, compared with all of the other variables at port. The wind map shows that the reduction of SO$_2$ was even more notable in almost every wind direction that blew from the navigational channel of JT to our observational site (Fig. 5). As shown in Fig. 1c, a westerly wind blew through the third pool, and a northeasterly wind blew through the second pool, and the SO$_2$ concentration dropped significantly in both directions. On the contrary, in the southwest direction, where wind blew from the city and road, SO$_2$ barely changed, indicating steady non-marine anthropogenic emissions. From this perspective, fuel switching in JT indeed resulted in a reduction in the ambient SO$_2$ concentration of around 70 %.

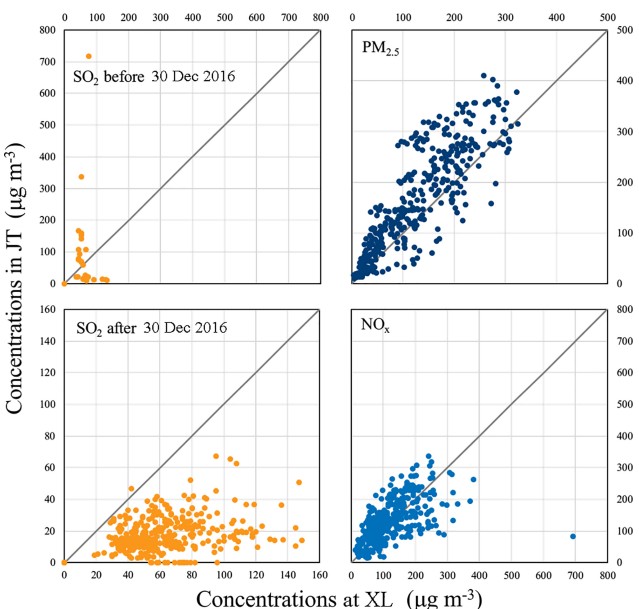

**Figure 4.** $NO_x$, $SO_2$ and $PM_{2.5}$ concentrations in Jingtang Harbor (JT) and at Xinli Primary School (XL).

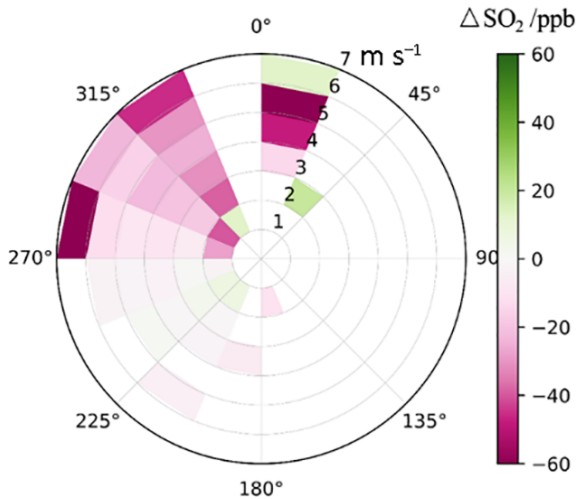

**Figure 5.** Distribution of differences in the $SO_2$ concentration by wind direction before and after 30 December 2016 in Jingtang Harbor. ($\triangle SO_2$ refers to $SO_2$ after 30 December 2016–$SO_2$ before 30 December 2016).

### 3.1.2 Carbonaceous and ionic species affected by marine vessels

Variations in carbonaceous and ionic species are depicted in Fig. 6. The mean (range of) concentrations of carbonaceous species determined in $PM_{2.5}$ were 6.52 (5.46–7.69) $\mu g\,m^{-3}$ for EC, and 23.10 (9.88–41.60) $\mu g\,m^{-3}$ for OC (OC levels are uncorrected for artifacts from absorption/volatilization of gaseous organic species). The levels of EC and OC were fairly consistent with that of $PM_{2.5}$ collected during the same

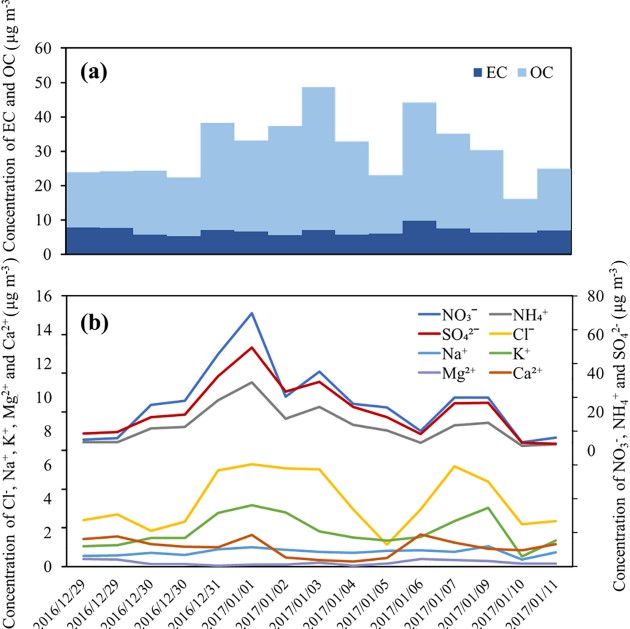

**Figure 6.** Variation of **(a)** carbonaceous species and **(b)** ionic species in $PM_{2.5}$.

period in Beijing (X. Li et al., 2018). EC is considered to be a tracer for primary emissions (incomplete combustion), from sources such as ships, vehicles and power plants in our study, which are affected by fuel quality and combustion. However, little variation was observed after fuel switching in this study due to the complicated contributors in the study are (JT). On the contrary, OC concentrations were much higher with large variation, showing the clear prevalence of organic carbonaceous species over EC. However, no discernible effect of fuel switching was found on OC concentrations.

As the major long-range transported aerosol components, $SO_4^{2-}$, $NO_3^-$ and $NH_4^+$ dominated the ionic species, with an average concentration of 22.04, 25.95 and 13.55 $\mu g\,m^{-3}$, respectively, and strong correlations with one another. $Ca^{2+}$, $Mg^{2+}$, $Na^+$, $K^+$ and $Cl^-$ are major constituents of sea salt and mineral dust, with an average concentration of 1.10, 0.21, 0.84, 2.10 and 3.90 $\mu g\,m^{-3}$, respectively (Fig. 6). Port-related emissions have been proven to be one of the major sources of local emission in JT. $Cl^-$ was relatively abundant, as the $Cl^-/Na^+$ ratio in the aerosol was 4.79, which is much larger than the ratio in the sea salt (1.8), indicating other strong anthropogenic sources such as coal combustion (Yao et al., 2002) and biomass burning (Li et al., 2007, 2009). $Ca^{2+}$, as an indicator of mineral dust, was higher than that observed in the city area of Tangshan (0.7 $\mu g\,m^{-3}$), which was considered to be related to port activities (load and unload bulk materials). Moreover, the mass ratio of $Mg^{2+}/Na^+$ (0.27) was higher than the value of 0.12 reported for sea water, suggesting additional magnesium sources such as soil dust containing dolomite, which may also be related to port

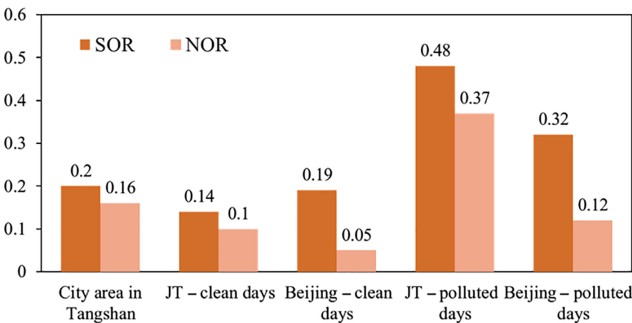

**Figure 7.** Sulfur oxidation rate (SOR) and the nitrogen oxidation rate (NOR) of particles collected in this study and in the city areas of Tangshan (Zhang et al., 2017) and Beijing (X. Li et al., 2018).

activities. To evaluate the contribution of stationary emissions and mobile emissions to air pollution (Gao et al., 2011), the mean mass ratio of $NO_3^-/SO_4^{2-}$ in JT was calculated; the result (1.19) was higher than that of that of the city area of Tangshan (0.7), indicating that JT was more affected by mobile sources than city area.

The high humidity in JT promotes secondary aerosol formation from local emissions (Yu et al., 2018). As the chemical composition of atmospheric particulate matter is largely affected by prevailing weather conditions, the samples were categorized as "polluted days" or "clean days" based on the corresponding $PM_{2.5}$ mass concentration for comparison (Röösli et al., 2001). The proportion of sulfates and nitrates rose from 55 % on clean days to 70 % on polluted days, showing a large concentration of secondary aerosols either due to transportation or formation. Generally, the SORs and NORs in JT were higher than those of city areas of Tangshan and Beijing (Fig. 7), and increased significantly from clean days (0.14 and 0.10, respectively) to polluted days (0.48 and 0.37, respectively), suggesting a strong localized photochemical redox reaction. The OC/EC ratio can be used as an indicator of the extent of the formation of secondary organic aerosols (Cabada et al., 2004). Despite the fact that the OC/EC emission ratio is dependent on both fuel type and engine type, tests show that it is still able to be utilized to distinguish between marine combustion sources (ratios typically over 10) (Celo et al., 2015; Moldanová et al., 2009; Sippula et al., 2014) and on-road diesel engines (ratios typically ranging from 0.25 to 1) (Oanh et al., 2010). In this study, the mean OC/EC ratio was 3.58, which is much higher than that of the Port of Thessaloniki in Greece (Tolis et al., 2015) and a port in Hong Kong (Gao et al., 2016), which indicates a higher influence from ship emissions in JT. Therefore, the localized photochemical reactions and aerosol formation driven by ship emissions contribute remarkably to air pollution in JT. This sheds light on the fact that secondary pollution should be treated via the reduction of local $SO_2$ emissions as part of pollution control in this harbor, which may be achieved by reducing ship emissions by fuel switching.

### 3.1.3 Elemental enrichment factors and marker for ship emissions

The ranges and mean concentrations of all measured elements are shown in Fig. 8. Overall, the mass concentrations of Al, Ti, Mg, Fe, Na, K, Mn, V, Ni, Zn and Pb were abundant and they varied largely with the sample time. Samples were categorized into three batches based on the $PM_{2.5}$ limit of Chinese IAQI level standard (HJ 633-2012) during sampling, considering the influences of ambient pollution on particulate chemical composition. One specific sample 2017/01/04 was set as a background/control group due to no ship activity during its sampling time (according to the ship traffic information provided), and intra-batch comparison was then performed to estimate the variations in elements after fuel switching.

The enrichment factor (EF) was used to normalize the observed concentrations of elements and to evaluate the influences of crustal and anthropogenic sources. Generally, elements from Ca to K in Fig. 8 are mainly from geological sources and are thus classified as "crustal elements". Ca was mostly from stable crustal sources which had the lowest EF values. With EF values around 10, and no evident temporal variation, the elements from Ti to U in Fig. 8 may have a major local crustal origin such as dust. Regression analysis comparing the EF values of Na and K revealed a strong correlation ($R^2$ value of 0.994), implying a primarily marine source. Conversely, Co, Mn and V were moderately enriched (EF < 100) and the elements from Ni to Se in Fig. 8 were highly enriched (EF > 100) due to the contribution of anthropogenic sources, and were all classified as "pollution elements". Co, Mn, Cu and Zn may have originated from the various bulk materials carried within the harbor area (Almeida et al., 2012; Moreno et al., 2007). With EF values that are strongly correlated with one another, Mn, TI, Cu, As, Sn, Zn, Pb, Hg and Ag would likely have a same major anthropogenic source, which is thought to be traffic pollution. According to tunnel studies (Lawrence et al., 2013; F. H. Li et al., 2018) and in situ measurements (Terzi et al., 2010; Thorpe and Harrison, 2008), Mn, Cu, Sn, Zn and Pb in $PM_{2.5}$ are related to vehicle emissions as well as tire and brake wear. Weckwerth (2001) reported As enrichment in $PM_{2.5}$ from the shaking of rusting rails due to passing trains; this would explain our observations, as our measurement site is in the vicinity of the train rails around the first pool. With respect to Mo and Cd, there were not enough values to present patterns, although the literature shows that Mo may be a contribution from diesel exhaust and brake wear (Weckwerth, 2001), whereas Cd may stem from motor vehicle emissions (F. H. Li et al., 2018). Again, this proved that JT was more affected by mobile sources, from which heavy metals generally originate.

The establishment of a marker to deduce variations in ship emissions is crucial. There has been a particular focus on Ni and V in $PM_{2.5}$, as several recent studies have clearly

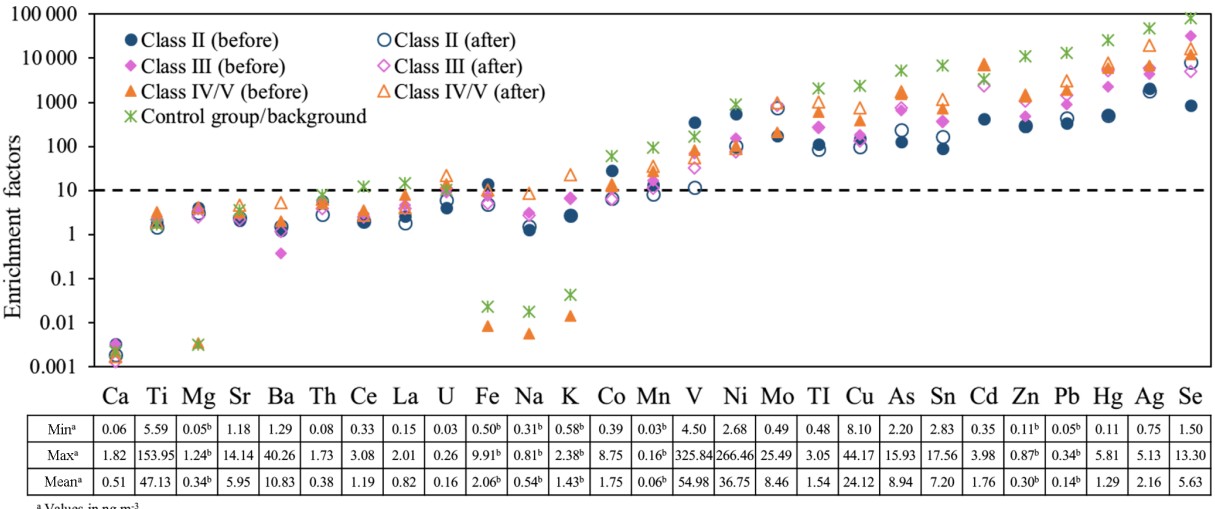

| | Ca | Ti | Mg | Sr | Ba | Th | Ce | La | U | Fe | Na | K | Co | Mn | V | Ni | Mo | Tl | Cu | As | Sn | Cd | Zn | Pb | Hg | Ag | Se |
|---|---|---|---|---|---|---|---|---|---|---|---|---|---|---|---|---|---|---|---|---|---|---|---|---|---|---|---|
| Min[a] | 0.06 | 5.59 | 0.05[b] | 1.18 | 1.29 | 0.08 | 0.33 | 0.15 | 0.03 | 0.50[b] | 0.31[b] | 0.58[b] | 0.39 | 0.03[b] | 4.50 | 2.68 | 0.49 | 0.48 | 8.10 | 2.20 | 2.83 | 0.35 | 0.11[b] | 0.05[b] | 0.11 | 0.75 | 1.50 |
| Max[a] | 1.82 | 153.95 | 1.24[b] | 14.14 | 40.26 | 1.73 | 3.08 | 2.01 | 0.26 | 9.91[b] | 0.81[b] | 2.38[b] | 8.75 | 0.16[b] | 325.84 | 266.46 | 25.49 | 3.05 | 44.17 | 15.93 | 17.56 | 3.98 | 0.87[b] | 0.34[b] | 5.81 | 5.13 | 13.30 |
| Mean[a] | 0.51 | 47.13 | 0.34[b] | 5.95 | 10.83 | 0.38 | 1.19 | 0.82 | 0.16 | 2.06[b] | 0.54[b] | 1.43[b] | 1.75 | 0.06[b] | 54.98 | 36.75 | 8.46 | 1.54 | 24.12 | 8.94 | 7.20 | 1.76 | 0.30[b] | 0.14[b] | 1.29 | 2.16 | 5.63 |

[a] Values in ng m$^{-3}$
[b] Values in μg m$^{-3}$, and mean (range) concentration of Al is 0.57 (0.08~1.75) μg m$^{-3}$

**Figure 8.** Enrichment factor of elements in PM$_{2.5}$ in Jingtang Harbor. The classes corresponding to the IAQI level standards computed from the PM$_{2.5}$ concentration during the sampling period. The mean, minimum and maximum concentrations of each element are also illustrated.

revealed that V and Ni are representative of ship exhaust particles using an aerosol time-of-flight mass spectrometer (Ault et al., 2010; Healy et al., 2009); furthermore, higher V levels in ship emissions have been found to be associated with residual fuel combustion instead of distillate fuel (Agrawal et al., 2008; Celo et al., 2015). In JT, V and Ni were considered to mainly originate from anthropogenic sources, whereas they were considered to be crustal elements in the city area of Tangshan. This indicated a unique contributor in JT, which was clearly ships consuming residual fuel. However in this study, Ni was proven to be less credible as a residual fuel marker, as the concentration of Ni was even inconsistent between parallel samples collected on the same day. On the contrary, the concentration of V, which showed significant intra-batch decreases in all three pollution levels, proved to be highly related to fuel switching; hence, V was identified as the perfect marker for residual fuel emissions in JT.

The chemical composition of MDOs/MGOs indicates that V is below the detection limit, but PM$_{2.5}$ samples presented the existence of V after fuel switching. This suggests that regionally transmitted V should not be overlooked. According to back trajectories, during the sampling period of sample 2017/01/04, when no ship activity existed, the air mass in control group moved into the Bohai Bay, before turning back to JT. This air mass was able to bring in particles containing a large amount of V from ships cruising in the sea that could still legally use IFO, verifying the influence of air transportation on particle content (Fig. 9). As this pathway would effectively influence the content of air mass, other trajectories were clustered into three typical types based on the transport pathways of air masses, each representing continental dominant, marine dominant and mixed airflows (which are

plotted in Fig. 9). The V concentration after fuel switching was 17.4 ng m$^{-3}$ under coastal airflows (marine and mixing airflow together), which is much higher than 9.0 ng m$^{-3}$ under continental airflows, indicating the significant effect of ship emissions on coastal areas. As shown in Fig. 9, samples that shared similar transport patterns from Mongolia–Inner Mongolia were compared to rule out the portion of transmitted V. The results showed that ships had switched fuel in advance and, most importantly, that the implementation of low-S$_F$ fuel reduced V from ships by 97.1 %, from 309.9 ng m$^{-3}$ before fuel switching to 9.1 ng m$^{-3}$ after its implementation.

## 3.2  $\Delta NO_x / \Delta SO_2$ ratios and the fuel type indicated in ship plumes

In this study, ship plume events were used for the surveillance of emissions and the fuel types utilized by passing ships (see Sect. 2.3). In total, 16 ship plume events were measured during this campaign, for which the molar $\Delta NO_x / \Delta SO_2$ ratios fell into the range of 0.92–17.89 (Table 1). The accuracy of the molar ratios were justified by proving that losses of $NO_x$ and $SO_2$ in the plumes during the transit time to the instrument were small. Potential losses of $NO_x$ include photolysis during the daytime (Makkonen et al., 2012); conversion to $NO_3$, $N_2O_5$ and subsequently $HNO_3$ at night (McLaren et al., 2010); and the heterogeneous conversion of $NO_2$ to HONO on the aqueous surface of the ocean (Wojtal et al., 2011). The furthest berth in the prevailing direction was 1.5 km from the measurement site, indicating a maximum plume transport time of 9 min at a wind speed of 2.7 m s$^{-1}$. Using this transport time and a $NO_x$ lifetime of 3.7 h measured in ship plumes (Beirle et al., 2004), we concluded a maximum potential loss of 4 % $NO_x$. Furthermore, the loss of $SO_2$ was attributed to heterogeneous re-

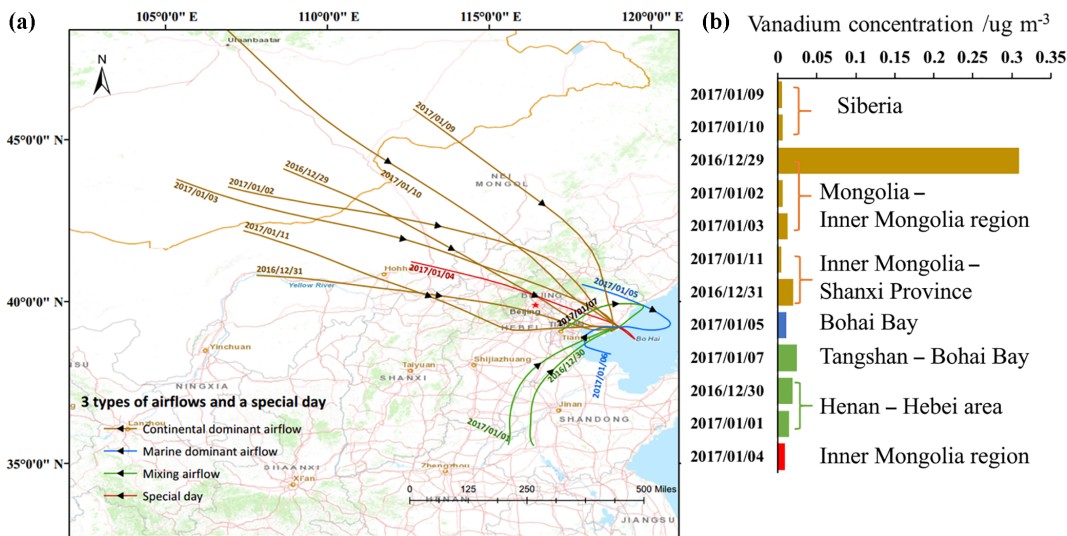

**Figure 9. (a)** Daily trajectories of air masses arriving at Jingtang Harbor during the sampling period, with starting dates listed next to the pathways. Sample 2017/01/04 is marked as a "special day" due to the absence of ship activity during its sampling period CE1. **(b)** The concentration of V in the PM$_{2.5}$ of each sample, clustered by origin and airflow type.

actions to form particle sulfate with equivalent lifetimes, which counteract the potential for NO$_x$ reactions to bias the $\triangle$NO$_x$/$\triangle$SO$_2$ ratios. Therefore, the maximum error in our measured $\triangle$NO$_x$/$\triangle$SO$_2$ ratios due to such loss processes is estimated to be $< 4\,\%$.

Previous inventories, measurements and ship plume studies have proven a direct correlation between S$_F$ and NO$_x$/SO$_2$ molar emission ratios, which aids with the determination of a critical point in this work. For high-S$_F$ fuels (2.43 %), several emission inventories for Bohai Bay indicated that the NO$_x$/SO$_2$ emission ratio was between 1.8 and 2.0 (Liu et al., 2016; Song, 2015; Xing et al., 2016), compared with the ratio of 2.6 observed from residual fuel plumes (Ault et al., 2010). For 0.5 % S$_F$ fuels in comparison, the inventory indicated 10.51 (Liu et al., 2016), which was also comparable with the ratio of 11.6 observed from distillate fuel plumes (Ault et al., 2010). The NO$_x$/SO$_2$ emission ratio rises as the S$_F$ value decreases, but it is also affected by the ship engine model, the load, the operation conditions and the combustion conditions in practical situations (McLaren et al., 2012). Thus, the variability in the observed $\triangle$NO$_x$/$\triangle$SO$_2$ ratios were expected, even when ships consumed the same type of fuel.

Considering all of the abovementioned aspects, we concluded that a ratio over 7.5 was a suggestion of fuel with a S$_F$ value below 0.5 %; therefore, ratios under 7.5 suggested high and incompliant S$_F$ values, as shown using the areas divided by the $x$ axis in Fig. 10. The $y$ axis in Fig. 10 represents the beginning of the ship fuel regulation within the three DECAs in the implementation plan. The axes make up four quadrants, each representing different scenarios. The ratios in first quadrant indicate the compliance of ship to the

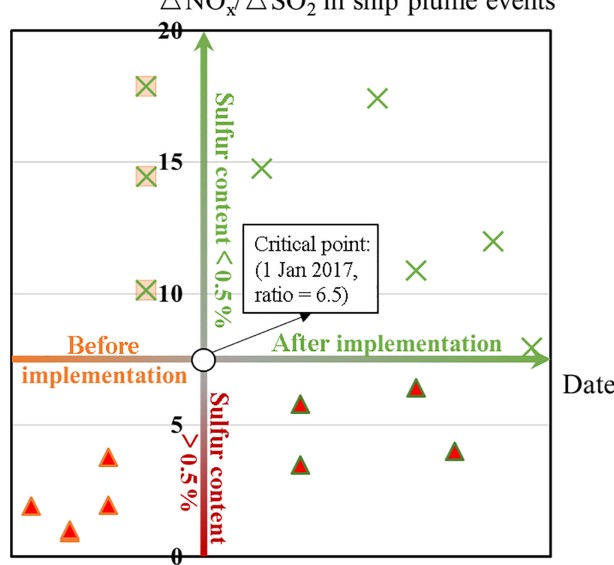

**Figure 10.** Molar NO$_x$/SO$_2$ ratios of all 16 ship plumes. The $x$ axis represents the date with the positive area referring to the time after 1 January 2017. The $y$ axis represents the NO$_x$/SO$_2$ ratio with the positive area referring to the S$_F$ < 0.5 % in fuel. Plumes within the four quadrants are distinguished using different symbols. TS1

regulation, and most of the ratios were higher than 10.51 – implying that the S$_F$ value was much lower than 0.5 %. Above the $x$ axis, ratios in the second quadrant also indicate compliance, in addition to the action of advance fuel switching before the implementation date. Ratios in the third quadrant (plume numbers 1-5) had an average $\triangle$NO$_x$/$\triangle$SO$_2$ ratio of 1.92, which conforms to the emission ratio in inven-

**Atmos. Chem. Phys., 19, 1–18, 2019**                                    www.atmos-chem-phys.net/19/1/2019/

tory (1.82 and 2.0) before the DECA implementation. Ratios in the fourth quadrant indicate the usage of high-$S_F$ fuel. As shown in Fig. 10, most of plumes indicate compliance with the 0.5 % $S_F$ limit, although some high-sulfur plumes still occurred. In these cases, precise identification of the high-sulfur plume contributors and the reinforcement of supervision are indeed necessary. Generally, the $SO_2$ reduction of the average $\Delta NO_x/\Delta SO_2$ ratio was 75 % from high-sulfur plumes (3.26) to low-sulfur plumes (12.97), which is consistent with the $S_F$ reduction (79 %) and the reduction of gas $SO_2$ in the ambient air (70 %, in Sect. 3.1.1); this proves the practicability of this method. One uncertain factor with respect to this method is the difficulty involved in identifying hybrid fuels that have a $S_F$ value of around 0.5 %. For $S_F$ values around 0.5 %, the $NO_x/SO_2$ emission ratio was observed to be either way below or consistent with the inventory estimate (around 6 in Winnes et al., 2016; around 11 in Zetterdahl et al., 2016), which may be attributed to the diversity of blending IFOs and MDOs. In this way, ships using hybrid fuels were unable to be identified, and some could be mistaken as incompliance. Further research is required on this subject.

## 3.3 Compliance based on the plumes

Test showed that the three ships we sampled from in 14 January 2017 burned MDOs (Table 2), which was in conformity with the implementation of the fuel regulation in JT. There would be obvious benefits such as significant improvements in emissions and air quality once all vessels comply and switch to MDOs or other alternative distillate fuels. Nevertheless, to enforce this, it is crucial to ensure the compliance of ships, which requires a more convenient and timely method of indicating fuel quality which does not involve analyzing fuel samples.

After identifying low-$S_F$ (compliant) and high-$S_F$ (potential incompliant) ship plumes, we matched each of the plumes with certain vessels using the ship traffic information, which contains a series of arrival and departure logs to help estimate the time when different ships passed the sampling site. Using the plume conditions, wind directions and ship traffic information to trace the specific source of measured plumes, we noted that most plumes were likely linked to several different ships because the wind often blew through the busy pools and navigational channel where many ships were manoeuvring and berthed. For the high-$S_F$ plume number 9, five ships were in berth in the upwind direction and two ships were passing by during the plume measurement period, which indicates a mixture of different individual plumes. A similar situation was found for other high-$S_F$ plumes: five ships were berthed and two were passing by during plume number 10; five ships were berthed and five were passing by during plume number 12; and four ships were berthed and four were passing by during plume number 14. In these cases, to achieve a comprehensive and accurate surveillance of the compliance of individual vessels, a more detailed and precise database of vessel activity, such as AIS data, is needed.

## 4 Conclusions and discussion

Field measurements were conducted at a measurement station in JT, including continued monitoring of meteorological conditions and gas and particle concentrations, from 28 December 2016 to 15 January 2017. Samples of $PM_{2.5}$ were collected every day from 28 December 2016 to 11 January 2017. Moreover, three fuel samples were taken from three respective vessels berthed in JT on 14 January 2017. Profiles of meteorological conditions and pollutants were obtained, in addition to the chemical characterization of aerosol and fuel samples.

Pollutant profiles showed a heavy polluted environment in JT in wintertime. On over 50 % of days, the $PM_{2.5}$ concentration was above the Chinese national ambient air quality standard class IV limit value (115 μg m$^{-3}$, China national standard GB 3095-2012). The average concentrations of $NO_x$, $SO_2$, $O_3$ and CO were 146.93, 21.91, 29.68 ppb and 2.21 ppm, respectively, among which $NO_x$ reached a maximum hourly concentration of 692.6 ppb and $SO_2$ reached a maximum hourly concentration of 165.5 ppb. Peak levels of $NO_x$ and $SO_2$ were mainly linked with ship activities, as the measurement site was very close to the channel and the berths, and a clear diurnal cycle of $O_3$ was noted due to changes in photochemical reactions and transportation. The mean (range of) concentrations of carbonaceous species in $PM_{2.5}$ were 6.52 (5.46–7.69) μg m$^{-3}$ for EC, and 23.10 (9.88–41.60) μg m$^{-3}$ for OC. $SO_4^{2-}$, $NO_3^-$ and $NH_4^+$ dominated the ionic species, with an average concentration of 22.04, 25.95 and 13.55 μg m$^{-3}$, respectively. $Ca^{2+}$, $Mg^{2+}$, $Na^+$, $K^+$ and $Cl^-$ were major constituents of sea salt and mineral dust, with an average concentration of 1.10, 0.21, 0.84, 2.10 and 3.90 μg m$^{-3}$, respectively. Enrichment factors of elements in $PM_{2.5}$ were used for the determination of a marker for residual fuel emissions, which was V in this study. Analyses of carbonaceous and ionic species revealed that local port-related emissions were one of the major sources of pollution in JT, especially the mobile sources. High humidity in port further exacerbated air pollution in the area by promoting localized photochemical reaction and secondary aerosol formation from ship emissions. Moreover, the effect of ship emissions were proven to be widespread because the concentration of V, the identified marker for residual fuel emissions, was much higher in coastal areas than continental areas.

After the implementation of low-sulfur fuel, fuel samples were collected from three vessels and were all found to be compliant with the fuel switching regulation. Based on previous studies and background at the measuring site, ship plume events were identified to be convenient for the surveillance of fuel quality. The $\Delta NO_x/\Delta SO_2$ ratios of all 16 ship plumes

fell within the range of 0.92–17.89, where which a ratio over 7.5 was identified as a suggestion of fuel with $S_F$ below 0.5 %, whereas values below 7.5 implied the use of fuel with a high and incompliant $S_F$ level. After the fuel switching implementation date, four plumes indicated the usage of high-$S_F$ fuel. However, compliance was difficult to conclude in these cases, and details and a precise database of the ships' locations would be required. Generally, the reduction of the average $\Delta NO_x / \Delta SO_2$ ratio from high-sulfur plumes (3.26) to low-sulfur plumes (12.97) shows a direct $SO_2$ emission reduction of 75 %, consistent with the $S_F$ reduction (79 %).

Despite the fact that carbonaceous species in particles were not significantly influenced by fuel switching, the gas and particle pollutants in the ambient air exhibited clear and effective improvements from the implementation of low-sulfur fuel. A comparison with the prevailing atmospheric conditions suggest a prompt 70 % $SO_2$ reduction in ambient air after 30 December 2016, which further analysis concluded to be a result of the reduction of local marine vessel sources. Given the high humidity at the site, this $SO_2$ reduction due to fuel switching will potentially abate the amount of secondary aerosols and improve the acidity of particulate matter in the region. As a marker for ship emission, the V concentration dropped by 97.1 % from 309.9 ng m$^{-3}$ before fuel switching to 9.1 ng m$^{-3}$ after, indicating a significant reduction due to the implementation of low-sulfur fuel.

*Data availability.* Data are available upon request.

*Supplement.* The supplement related to this article is available online at: https://doi.org/10.5194/acp-19-1-2019-supplement. TS2

*Author contributions.* YZ, FD, and HM contributed equally. YZ primarily participated in the chemical analyses and wrote the article. FD and HM mainly participated in the design and conducted the field measurements, which was considered to be an equal contribution to this work. MF helped design the experiments and was responsible for the pilot preparations. ZL and QX helped conduct the field measurements. XJ and SL contributed to setting instruments. KH provided constructive comments on this research. HL conceived this study and provided guidance on the whole research process as well revising the paper.

*Competing interests.* The authors declare that they have no conflict of interest.

*Special issue statement.* This article is part of the special issue "Shipping and the Environment – From Regional to Global Perspectives (ACP/OS inter-journal SI)". It is a result of the Shipping and the Environment – From Regional to Global Perspectives, Gothenburg, Sweden, 23–24 October 2017.

*Acknowledgements.* This work was supported by the National Science Fund for Excellent Young Scholars (grant no. 41822505), the National Natural Science Found of China (grant nos. 91544110 and 41571447), Beijing Nova Program (grant no. Z181100006218077), the National Key R&D Program (grant no. 2016YFC0201504), the Special Fund of State Key Joint Laboratory of Environment Simulation and Pollution Control (grant no. 16Y02ESPCT), the National Research Program for Key Issues in Air Pollution Control (grant no. DQGG0201&0207), and the National Program on Key Basic Research Project (grant no. 2014CB441301). We appreciate that Hebei Sailhero Environmental Protection High-tech Co., Ltd. and Guangzhou Hexin Instrument Co., Ltd. provided the instruments used for our observations. We are also grateful for all of the help from the Sino-Japan Friendship Centre for Environmental Protection and Sinopec Research Institute of Petroleum Processing.

*Review statement.* This paper was edited by Markus Quante and reviewed by two anonymous referees.

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

**Remarks from the language copy-editor**

CE1    Please note the slight changes to the requested edit.

**Remarks from the typesetter**

TS1    Thank you very much for the new file. While comparing the old and the new version of Fig. 10 we noticed that further changes have been made to the figure (e.g. "ratio = 6.5" is missing, legend has been added). Could you please check the changes again carefully and, additionally, correct the capitalization of the words (see current file)? Thank you very much in advance.

TS2    Please note that the deletion of the supplement at this stage should be approved by the editor. Please provide a short explanation that could by forwarded by us to the editor.