# Peer review of "Compliance and port air quality features of ship fuel switching regulation: by a field observation SEISO-Bohai"

_Atmospheric Chemistry and Physics, 2018_

## Referee Comment (RC1) · Anonymous Referee #1 · 13 Feb 2019

The manuscript presents results being very relevant considering the ongoing processes with ECA focus different emissions. The study is generally well written and well performed and well worth publishing. I think complementary information concerning the impact of emissions on water quality should be added and also that the fact that atmpospheric conditions has such a strong impact on the atmospheric concentrations could be further stresses. There are some typos and language corrections needed.

Some specific comments: Introduction: The manuscript focuses on air quality, but I suggest also to mention potential problems of shipping for water quality (for example according to Turner et al), as the two problems are related.

[Figure]

Page 3 Line 31: observations on a roof Page 4 line 4: what is "mismeasurements"? Page 7 eq 1: Difficult to see what is in the equation.

Turner DR, Hassellöv I-M, Ytreberg E, Rutgersson A. Shipping and the environment: Smokestack emissions, scrubbers and unregulated oceanic consequences. Elem Sci Anth. 2017;5:45. DOI: http://doi.org/10.1525/elementa.167

Turner, D. R, M. Edman, J. A. Gallego-Urrea, B. Claremar, I.-M. Hassellöv, A. Omstedt and A. Rutgersson (2018). The potential future contribution of shipping to acidification of the Baltic Sea. Ambio, DOI 10.1007/s13280-017-0950-6

---

## Referee Comment (RC2) · Anonymous Referee #2 · 18 Feb 2019

The issues raised in this paper are interesting and thoroughly approached from a broad perspective. The authors present a lot of new interesting results that is of high value to other researchers and in the long run to policy makers. I have one main concern on the presentation. I think the suggested method to use of the ratio SO2/NOX in order to determine whether a ship uses heavy fuel oil or distillate oil needs more discussion. NOX-emissions are mainly related to engine and combustion characteristics. My question is why not use SO2/CO2 ratio? Please add a discussion on that. I would also add that the manuscript needs a language check before publication.

Some details:

[Figure]

Rather specify sea areas than refer to emissions from ships in Europe, (page2, row 6) and if possible consider multiple references to the emission estimates. Similar comment to statement on row 14 on ship emissions in eastern China.

Page 2 row 24, The NOX reductions should not be confused to be accomplished by the fuel switch.

Page 4 row 10 on hourly measurements of PM2.5 and PM10 by $\beta$-ray absorption should be explained. This is probably not the $\mu$g/m3 measurements.

Suggest to change "aerosol sample" to something like e.g. "exposed filter"

Explain more on why only 16 plume events are identified - the period was long and the port is described as very busy.

page 6 row 16. Suggest rewrite "In addition, high concentrations of organics, metals and the compounds between are obtained in IFOs from their presence in the original crude oil." This is an unclear statement.

Page 6 on hybrid fuels: It is important to point out that these fuels can be anything from low sulphur heavy oils to qualities close to gasoils. The important issue is that there is no standard for these fuels (e.g. ISO-standard) and the only rquirement is that the sulphur is less than specified (<0.1%)

Page 10 row 7. The OC/EC ratio in ship emissions is probably both dependent to fuel (residual or distillate) and to engine characteristics and therefore varies a lot.

Figure 8. There should be an explanation to what is meant by the different "classes" in the Figure caption.

---

## Author Comment (AC1) · 4 Mar 2019

Please see the supplement for formatted responses.

Response to Referee's Comments #1 1. In the Introduction section: It is suggested to mention potential problems of shipping for water quality (for example according to Turner et al), as the two problems are related. Response: Thanks a lot for your suggestion and recommendation. We surely agree with you that the impact of ship emissions on water quality is not negligible. It is worth mentioning that some emission reduction methods such as scrubbers, could cause the pollution of surface water. And adding it into the introduction section, makes a more comprehensive understanding for readers

of how we should properly reduce the air pollution from ship emissions instead of shifting it to water. So we summarize some relative research and revise the manuscript as shown below. Revision in manuscript: Page 2, Line 23-26: These situations have constantly drawn attention on coastal air pollution and correlative emission control strategy such as scrubbers. However, recent research also presents the potential pollution of ship emissions to surface water due to some methods of treating ship exhausts (Hassellöv et al., 2013; Stips et al., 2016; Turner et al., 2018; Turner et al., 2017), which reminds us to be more careful about ship emission reduction.

2. There are some typos and language corrections needed. For example: Page 3 Line 31: observations on a roof; Page 4 line 4: what is "mismeasurements"? ; Page 7 eq 1: Difficult to see what is in the equation. Response: Thanks for the suggestion. We check through the manuscript and all the language and typo errors are corrected. The manuscript does have an inaccurate description "mismeasurement" which was to describe that the data like that was wrong. We have it revised as shown below. The equation was a little difficult to see as it was restricted by specific format requirement that makes some parts even smaller in display. So we change the display style for the fraction and make the equation more evident as shown below. Revision in manuscript: (eight examples) Page 4, Line 13-15: A small meteorological monitoring station was placed on the roof of the container and obtained temperature (℃), relative humidity (%), wind speed (m•s-1), wind direction and radiation intensity every 1 min, from 28 December 2016 to 15 January 2017. Page 4, Line 15-16: Abrupt high temperature values were subtracted from results because they were obvious invalid data when instrument indicated 40℃ for ambient temperature in winter. Page 4, Line 24-25: Invalid values of O3 occurred fitfully during the campaign, appearing as a sinusoid fluctuation below 10 ppb, which were subtracted from the results. Page 8, eq. 1: EF=(X/R)aerosol/(X/R)crust Page 8, eq. 2: SOR= [SO_4ˆ(2-) ]/(([SO_4ˆ(2-) ]+[SO_2 ]) ) Page 8, eq. 3: NOR= [NO_3ˆ- ]/(([NO_3ˆ- ]+[NO_2 ]) ) Page 9, Line 23-24: Peak levels of NOx and SO2 were mainly linked with ship activities since the measurement site was very close to channel and berth. Page 9, Line 25-26: A clear diurnal cycle of

O3 was spotted that the concentration rises in daytime (29.18 ppb) and falls at night (16.38 ppb).

Reference Hassellöv, I.-M., R. Turner, D., Lauer, A., and Corbett, J.: Shipping contributes to ocean acidification, 2731-2736 pp., 2013. Stips, A., Bolding, K., Macías, D., Bruggeman, J., and Eayrs, C.: Scoping report on the potential impact of on-board desulphurisation on the water quality in SOx Emission Control, 2016. Turner, D. R., Hassellöv, I.-M., Ytreberg, E., and Rutgersson, A.: Shipping and the environment: Smokestack emissions, scrubbers and unregulated oceanic consequences, 45 pp., 2017. Turner, D. R., Edman, M., Gallego Urrea, J., Claremar, B., Hassellöv, I.-M., Omstedt, A., and Rutgersson, A.: The potential future contribution of shipping to acidification of the Baltic Sea, 2018.

Please also note the supplement to this comment:
https://www.atmos-chem-phys-discuss.net/acp-2018-1233/acp-2018-1233-AC1-supplement.pdf

---

## Author Comment (AC2) · 4 Mar 2019

Please see the supplement file for the formatted responses. The plain text was provided here.

Response to Referee's Comments 2

1. The ratio $SO_2/NO_X$ in order to determine whether a ship uses heavy fuel oil or distillate oil needs more discussion. $NO_X$-emissions are mainly related to engine and combustion characteristics. My question is why not use $SO_2/CO_2$ ratio?

Response:

Thanks for your suggestion. We agree that the SO2/CO2 ratio of ship plume is a better indicator of fuel sulfur content than the SO2/NOx ratio, and is widely used in several in site measurements (Kattner et al., 2015; Loov et al., 2014; Yang et al., 2016). However, we are aware of that in China, the concentration of CO2 is excluded from ambient air measurement. The Ambient Air Quality Standards (China National Standard GB 3095-2012) stipulates the monitoring of six pollutants in ambient air, which is SO2, NO2 (or NOx), O3, CO, PM2.5 and PM10. Hence, the CO2 measurement is not equipped on from thousands of micro-monitor stations to medium or even mobile monitor stations. To use monitoring capacity maximumly, we would like to explore a relatively reliable and practical indicator within the six pollutants mentioned above. According to the Third IMO GHG Study, NOx emissions do vary by engine and combustion. But based on the ship information provided by JT, the size of berth and the design of port, we found ships in JT, especially those related to identified plumes, mainly consistent in size. Actually, ship information provided by the port during campaign indicates the variance of ship size is little in Jingtang Harbor. All these conditions make the SO2/NOx ratio reliable and convincing for indicating fuel sulfur content in Jingtang Harbor. We believe it would be more appropriate to use SO2/CO2 ratio, but with the absence of CO2 concentration and the consistency of ship size, the ratio SO2/NOx is also applicable.

Revision in manuscript:

Page 6, Line 12-15: The SO2 to CO2 ratio in ship plume is widely used as an indicator for SF (Yang et al., 2016; Kattner et al., 2015; Loov et al., 2014). However, in this study we intend to explore another applicable indicator for situation in China that the concentration of CO2 is often excluded from ambient air measurement after the Ambient Air Quality Standards (China National Standard GB 3095-2012) stipulates the six pollutants to monitor without CO2. Page 6, Line 17-20: Moreover, based on the ship information provided by JT, the size of berth and the design of port, we found ships in JT, especially those related to identified plumes, mainly consistent in size, which implies similar NOx emissions in those plumes (IMO, 2015). Therefore the NOx to SO2

ratio is appropriate to indicate the SF of ships in JT.

2. Rather specify sea areas than refer to emissions from ships in Europe, (page 2, row 6) and if possible consider multiple references to the emission estimates. Similar comment to statement on row 14 on ship emissions in eastern China.

Response:

Thanks for your suggestion. We tried to describe the impact of ship emissions in Europe and eastern China to lay a background for their corresponding ship emission regulation. And we agree with you that the statement was a little vague about sea areas and that our statement needs more references. So we specify the sea areas and summarize some valuable works as revision below.

Revision in manuscript:

Page 2, Line 6-11: In the EU-27, ships in 2005 emitted 2.8 million tons NOx, 1.7 million tons SO2 and 0.2 million tons PM2.5, of which approximately 70 Page 2, Line 18-22: Estimation of ship emissions within 200 nm to the Chinese coast showed that ship emissions accounted for an annual increase of up to 5.2 $\mu$g•m-3 PM2.5 in eastern China, which influenced the air quality in not only coastal areas but also the inland areas hundreds of kilometers away from the sea (Lv et al., 2018). In 2010, ships contributed 12.0

3. Page 2 row 24, The NOX reductions should not be confused to be accomplished by the fuel switch.

Response:

Thanks for your suggestion. This paragraph was to present the expectation and effect of the fuel switch, and the NOx reductions mentioned here was confused just as you suggested. So we get rid of the NOx reductions in the manuscript.

Revision in manuscript:

Page 2, Line 32-33: Estimation shows that IMO limitation of 0.1

4. Page 4 row 10 on hourly measurements of PM2.5 and PM10 by $\beta$-ray absorption should be explained. This is probably not the $\mu$g/m3 measurements.

Response:

Thanks for your suggestion. According to the International Organization for Standardization, the $\beta$-ray absorption method is a method for the measurement of the mass of particulate matter in ambient air and is based on the absorption of beta rays by the particulate matter (ISO 10473:2000). The concentration was computed as follows: $C_{p}articles$ = ($\Delta$m·S(Detection area,cm$^2$))/($q(sampling flow, m^3/h)$·t(sampling time,h)) , where the mass per unit area of the particulate matter trapped in the filter $\Delta$m(mg/cm$^2$) = $ln(N_1/N_2)/(k(absorption coefficient, cm^2/mg)) The N1 and N2 represent the amount of \beta$-ray passing through a blank filter and that trapped by particulate matter, respectively. Similarly, several research describing the $\beta$-ray absorption method just as the equation above (Zhao et al., 2013; Zuo et al., 2017).

Revision in manuscript:

Page 4, Line 20-22: Monitoring modules consist of NO, NO2 and NOx measurement by an analyser, SO2 detection by UV fluorometric, CO by IR absorption, O3 by UV spectrophotometry, and particles by $\beta$-ray absorption (ISO 10473:2000).

5. Suggest to change "aerosol sample" to something like e.g. "exposed filter"

Response:

Thanks for your suggestion. We agree that "aerosol sample" is not as equally accurate as "exposed filter". And we revise the manuscript as below.

Revision in manuscript:

Page 5, Line 1: 2.1.4 Particle samples Page 5, Line 3-5: The filters were exposed for

23 h (normally from 16:30 to 15:30 LST the next day, local standard time, and named after the ending date) on an 80 mm-diameter pre-fired quartz microfiber filters (CHM QF1 grade) by a Laoying Model 2030 TSP sampler. Page 5, Line 10-12: 0.55 cm2 section of each exposed filter and blank filters were measured for concentrations of organic carbon (OC) and elemental carbon (EC) by the Thermal Optical Transmission Method in a DRI 2001 organic carbon/elemental carbon (OC/EC) analyser. Page 5, Line 19-20: 50 cm2 section of each exposed filter and blank filters were extracted with 10 ml ultra-pure water in an ultrasonic bath at 4 âĐČ for 30 min. Page 5, Line 25-26: 20 cm2 section of each exposed filter and blank filters were digested with 25 ml of an 8

6. Explain more on why only 16 plume events are identified - the period was long and the port is described as very busy.

Response:

Thanks for your suggestion. The port is busy indeed, but there are several reasons for the rather few plume event. Firstly, during our measurement, there was a period when ships barely went into the port due to the New Year holidays and also due to the poor visibility from January 1 to 4, 2017. Secondly, the wind direction in JT changes quickly, and sometimes it was unfavorable for instrument to capture the ship plume. And the prevailing wind direction indicates our plumes would be mainly from the 2nd pool and the 3rd pool, of which approximately half berth were actually in construction and not in use, making our plumes quite few. Thirdly, the port is actually quite polluted (in over 50

Revision in manuscript:

Page 6, Line 16-31: For these time stamps, peaks in NOx along with simultaneous valleys in O3 were then identified in valid data. The signals were only affirmed when there were significant peaks and clearly determinable backgrounds. Finally ship plume event were marked if the existence of ships was positive in the upwind direction of those signals. The combination of the trace gas peak time, the wind direction, and

the ship traffic information (time of ships leaving and berthing) provided by marine administration in the port will enable the identification of the plume-related ship. For example, a ship plume event was identified in 5 January 2017 from 15:36 to 16:08 (Fig. 2). The timing and conditions associated with 16 positively identified ship plume event are listed in Table 1. Several situations made it more difficult to identify a ship plume event in our measurement. Firstly, there was a period when ships barely went into the port due to the New Year holidays and also due to the poor visibility from January 1 to 4, 2017. Secondly, the prevailing wind direction indicates our plumes would be mainly from the 2nd pool and the 3rd pool, of which approximately half berth were actually in construction and not in use, making our plumes quite fewer than expect, let alone the fact that wind direction is actually changes quickly and sometimes unfavourable for instrument to capture the ship plume. Thirdly, the port is actually quite polluted (in over 50

7. Page 6 row 16. Suggest rewrite "In addition, high concentrations of organics, metals and the compounds between are obtained in IFOs from their presence in the original crude oil." This is an unclear statement.

Response:

Thanks for your suggestion. We agree with you that the statement is unclear and can be quite confusing. The manuscript is revised as below.

Revision in manuscript:

Page 7, Line 17-18: In addition, IFOs obtain high concentrations of organics, metals, and the compounds of organic metal from their presence in the original crude oil.

8. Page 6 on hybrid fuels: It is important to point out that these fuels can be anything from low sulphur heavy oils to qualities close to gasoils. The important issue is that there is no standard for these fuels (e.g. ISO-standard) and the only rquirement is that the sulphur is less than specified (<0.1

Response:

Thanks for your suggestion. We agree with you that the situation of hybrid fuels should be stated clearly and the previous description is quite vague. We revise the manuscript and point out the unsupervised situation of hybrid fuels.

Revision in manuscript:

Page 7, Line 31-Page 8, Line 3: Another record worth mentioning is that hybrid fuels that blend IFO and other low SF fuels to comply with SF limit are found widely used by ships operating in SECAs (Winnes et al., 2016; Zetterdahl et al., 2016), since the price of distillate fuels is an obstacle for contractors to completely abandon IFOs. However, by now ISO 8217:2017, the benchmark for the quality of marine fuels on the market, has not obtain any limits of physical and chemical parameters for hybrid fuels. It causes a large uncertainty of their qualities since there are zero formal standard for quality of hybrid fuels except the requirement of SF.

9. Page 10 row 7. The OC/EC ratio in ship emissions is probably both dependent to fuel (residual or distillate) and to engine characteristics and therefore varies a lot.

Response:

Thanks for your suggestion. The OC/EC ratio is indeed under influence of both fuel and engine, but several emission factor studies suggest the fact that OC/EC emission ratio is strongly distinguishable between marine combustion (typically over 10) and on-road diesel engine (typically lower than 1) (Celo et al., 2015; Khan et al., 2012; Moldanová et al., 2009; Oanh et al., 2010; Sippula et al., 2014). Therefore, the higher value of OC/EC ratio of aerosols in JT may indicate the worse influence of ship emissions than other port city like Hong Kong. We revise the manuscript with a more appropriate expression as below.

Revision in manuscript:

Page 11, Line 6-10: Despite the OC/EC emission ratio dependent to both fuel type and

engine, tests show that it is still strongly distinguishable between marine combustions (typically over 10) (Celo et al., 2015; Moldanová et al., 2009; Sippula et al., 2014) and on-road diesel engine (typically ranging from 0.25 to 1) (Oanh et al., 2010). In this study, the mean OC/EC ratio was 3.58, much higher than that of Thessaloniki port in Greece and Hong Kong, which indicates a worse influence of ship emissions in JT.

10. Figure 8. There should be an explanation to what is meant by the different "classes" in the Figure caption.

Response:

Thanks for your suggestion. The categorization of "classes" are described in page 10 line 15 as "Samples were categorized into three batches based on the PM2.5 limit of AQI level (HJ 633-2012) during sampling, considering the influences of ambient pollution on particulate chemical composition". And for convenience of readers, we add the explanation in the title of Figure 8.

Revision in manuscript:

Figure 8: Enrichment factor of elements in PM2.5 in JingTang Harbor. The classes are corresponding AQI level computed from PM2.5 concentration during sampling time. The mean, minimum, and maximum concentrations of each element are also illustrated.

Reference

Campling, P., Janssen, L., Vanherle, K., Cofala, J., Heyes, C., and Sander, R.: Final Report: Specific evaluation of emissions from shipping including assessment for the establishment of possible new emission control areas in European Seas, The Flemish Institute for Technological Research NV, 74, 2013.

Celo, V., Dabek-Zlotorzynska, E., and McCurdy, M.: Chemical Characterization of Exhaust Emissions from Selected Canadian Marine Vessels: The Case of Trace Metals and Lanthanoids, Environmental Science Technology, 49, 5220-5226,

10.1021/acs.est.5b00127, 2015.

Fu, Q., Shen, Y., and Zhang, J.: On the ship pollutant emission inventory in Shanghai port, Journal of Safety and Environment, 12, 57-64, 2012.

IMO: Third IMO GHG Study 2014, 2015.

ISO 10473:2000, Ambient air — Measurement of the mass of particulate matter on a filter medium — Beta-ray absorption method.

ISO 8217:2017. Petroleum products-Fuels (class F)-Specifications of marine fuels.

Jalkanen, J. P., Johansson, L., and Kukkonen, J.: A comprehensive inventory of the ship traffic exhaust emissions in the Baltic Sea from 2006 to 2009, Ambio, 43, 311-324, 10.1007/s13280-013-0389-3, 2014.

Jalkanen, J. P., Johansson, L., and Kukkonen, J.: A comprehensive inventory of ship traffic exhaust emissions in the European sea areas in 2011, Atmospheric Chemistry and Physics, 16, 71-84, 10.5194/acp-16-71-2016, 2016.

Khan, M. Y., Giordano, M., Gutierrez, J., Welch, W. A., Asa-Awuku, A., Miller, J. W., Cocker, D. R. (2012). Benefits of Two Mitigation Strategies for Container Vessels: Cleaner Engines and Cleaner Fuels. ENVIRONMENTAL SCIENCE TECHNOLOGY, 46(9), 5049-5056. doi:10.1021/es2043646 Kattner, L., Mathieu-Uffing, B., Burrows, J. P., Richter, A., Schmolke, S., Seyler, A., Wittrock, F. (2015). Monitoring compliance with sulfur content regulations of shipping fuel by in situ measurements of ship emissions. Atmospheric Chemistry and Physics, 15(17), 10087-10092. doi:10.5194/acp-15-10087-2015

Li, C., Yuan, Z., Ou, J., Fan, X., Ye, S., Xiao, T., Shi, Y., Huang, Z., Ng, S. K. W., Zhong, Z., and Zheng, J.: An AIS-based high-resolution ship emission inventory and its uncertainty in Pearl River Delta region, China, Science of the Total Environment, 573, 1-10, 10.1016/j.scitotenv.2016.07.219, 2016.

Loov, J. M. B., Alfoldy, B., Gast, L. F. L., Hjorth, J., Lagler, F., Mellqvist, J., . . . Borowiak, A. (2014). Field test of available methods to measure remotely SOx and NOx emissions from ships. Atmospheric Measurement Techniques, 7(8), 2597-2613. doi:10.5194/amt-7-2597-2014.

Lv, Z., Liu, H., Ying, Q., Fu, M., Meng, Z., Wang, Y., Wei, W., Gong, H., and He, K.: Impacts of shipping emissions on PM2.5 air pollution in China, Atmospheric Chemistry and Physics Discussion, https://doi.org/10.5194/acp-2018-540, 2018.

Moldanová, J., Fridell, E., Popovicheva, O., Demirdjian, B., Tishkova, V., Faccinetto, A., and Focsa, C.: Characterisation of particulate matter and gaseous emissions from a large ship diesel engine, Atmospheric Environment, 43, 2632-2641, 10.1016/j.atmosenv.2009.02.008, 2009.

Oanh, N. T. K., Thiansathit, W., Bond, T. C., Subramanian, R., Winijkul, E., and Pawarmart, I.: Compositional characterization of PM2.5 emitted from in-use diesel vehicles, Atmospheric Environment, 44, 15-22, 10.1016/j.atmosenv.2009.10.005, 2010.

Sippula, O., Stengel, B., Sklorz, M., Streibel, T., Rabe, R., Orasche, J., Lintelmann, J., Michalke, B., Abbaszade, G., Radischat, C., Groeger, T., Schnelle-Kreis, J., Harndorf, H., and Zimmermann, R.: Particle Emissions from a Marine Engine: Chemical Composition and Aromatic Emission Profiles under Various Operating Conditions, Environmental Science Technology, 48, 11721-11729, 10.1021/es502484z, 2014.

Winnes, H., Moldanova, J., Anderson, M., and Fridell, E.: On-board measurements of particle emissions from marine engines using fuels with different sulphur content, Proceedings of the Institution of Mechanical Engineers Part M-Journal of Engineering for the Maritime Environment, 230, 45-54, 10.1177/1475090214530877, 2016.

Yang, M., Bell, T. G., Hopkins, F. E., Smyth, T. J. (2016). Attribution of atmospheric sulfur dioxide over the English Channel to dimethyl sulfide and changing ship emissions. Atmospheric Chemistry and Physics, 16(8), 4771-4783. doi:10.5194/acp-16-

4771-2016.

Zetterdahl, M., Moldanova, J., Pei, X., Pathak, R. K., and Demirdjian, B.: Impact of the 0.1

Zhao, X., Pan, J. X., Liu, B., and Chen, P.: Research of measuring technology for pm2.5 based on the $\beta$-ray absorption method. Application of Electronic Technique, 2013. (in Chinese)

Zuo, J. and Zhao Z.: Research of PM2.5 Monitoring Key Technology Based on $\beta$-Ray Method. Instrument Technique and Sensor, 2017. (in Chinese)

Please also note the supplement to this comment:
https://www.atmos-chem-phys-discuss.net/acp-2018-1233/acp-2018-1233-AC2-supplement.pdf

**Supplement:**

**Response to Referee's Comments #2**

1. The ratio $SO_2/NO_X$ in order to determine whether a ship uses heavy fuel oil or distillate oil needs more discussion. $NO_X$-emissions are mainly related to engine and combustion characteristics. My question is why not use $SO_2/CO_2$ ratio?

5 **Response:**

Thanks for your suggestion. We agree that the $SO_2/CO_2$ ratio of ship plume is a better indicator of fuel sulfur content than the $SO_2/NO_x$ ratio, and is widely used in several in site measurements (Kattner et al., 2015; Loov et al., 2014; Yang et al., 2016). However, we are aware of that in China, the concentration of $CO_2$ is excluded from ambient air measurement. The Ambient Air Quality
10 Standards (China National Standard GB 3095-2012) stipulates the monitoring of six pollutants in ambient air, which is $SO_2$, $NO_2$ (or $NO_x$), $O_3$, CO, $PM_{2.5}$ and $PM_{10}$. Hence, the $CO_2$ measurement is not equipped on from thousands of micro-monitor stations to medium or even mobile monitor stations. To use monitoring capacity maximumly, we would like to explore a relatively reliable and practical indicator within the six pollutants mentioned above. According to the Third IMO GHG
15 Study, $NO_x$ emissions do vary by engine and combustion. But based on the ship information provided by JT, the size of berth and the design of port, we found ships in JT, especially those related to identified plumes, mainly consistent in size. Actually, ship information provided by the port during campaign indicates the variance of ship size is little in Jingtang Harbor. All these conditions make the $SO_2/NO_x$ ratio reliable and convincing for indicating fuel sulfur content in
20 Jingtang Harbor. We believe it would be more appropriate to use $SO_2/CO_2$ ratio, but with the absence of $CO_2$ concentration and the consistency of ship size, the ratio $SO_2/NO_x$ is also applicable.

**Revision in manuscript:**

1) *Page 6, Line 12-15: The $SO_2$ to $CO_2$ ratio in ship plume is widely used as an indicator for $S_F$*
25 *(Yang et al., 2016; Kattner et al., 2015; Loov et al., 2014). However, in this study we intend to explore another applicable indicator for situation in China that the concentration of $CO_2$ is often excluded from ambient air measurement after the Ambient Air Quality Standards (China National Standard GB 3095-2012) stipulates the six pollutants to monitor without $CO_2$.*

2) *Page 6, Line 17-20: Moreover, based on the ship information provided by JT, the size of berth*
30 *and the design of port, we found ships in JT, especially those related to identified plumes, mainly consistent in size, which implies similar $NO_x$ emissions in those plumes (IMO, 2015). Therefore the $NO_x$ to $SO_2$ ratio is appropriate to indicate the SF of ships in JT.*

2. Rather specify sea areas than refer to emissions from ships in Europe, (page 2, row 6) and if

possible consider multiple references to the emission estimates. Similar comment to statement on row 14 on ship emissions in eastern China.

**Response:**

Thanks for your suggestion. We tried to describe the impact of ship emissions in Europe and eastern China to lay a background for their corresponding ship emission regulation. And we agree with you that the statement was a little vague about sea areas and that our statement needs more references. So we specify the sea areas and summarize some valuable works as revision below.

**Revision in manuscript:**

1) *Page 2, Line 6-11: In the EU-27, ships in 2005 emitted 2.8 million tons $NO_x$, 1.7 million tons $SO_2$ and 0.2 million tons $PM_{2.5}$, of which approximately 70 % occurred within 200 nm from the coast of EU Member States (Campling et al., 2013). From 2006 to 2009, $NO_x$ emission from ships rose by approximately 7 % in Baltic Sea, while $SO_2$ and $PM_{2.5}$ emissions reduced by 14 % and 20 %, respectively, mainly caused by the fuel requirements which became effective in 2006 (Jalkanen et al., 2014). In 2011, ship emission in Europe was estimated to be 2.0 million tons $NO_x$, 1.2 million tons $SO_2$ and 0.2 million tons $PM_{2.5}$ (Jalkanen et al., 2016).*

2) *Page 2, Line 18-22: Estimation of ship emissions within 200 nm to the Chinese coast showed that ship emissions accounted for an annual increase of up to 5.2 $\mu g \cdot m^{-3}$ $PM_{2.5}$ in eastern China, which influenced the air quality in not only coastal areas but also the inland areas hundreds of kilometers away from the sea (Lv et al., 2018). In 2010, ships contributed 12.0 % of $SO_x$, 9.0 % of $NO_x$ and 5.3 % of $PM_{2.5}$ in total emission in Shanghai (Fu et al., 2012). And it was obtained that 14.1 % of $SO_2$, 11.6 % of $NO_x$ and 3.6 % of $PM_{2.5}$ emission within the Pearl River Region, China in 2013, was attributed to ships (Li et al., 2016).*

3. Page 2 row 24, The NOX reductions should not be confused to be accomplished by the fuel switch.

**Response:**

Thanks for your suggestion. This paragraph was to present the expectation and effect of the fuel switch, and the NOx reductions mentioned here was confused just as you suggested. So we get rid of the NOx reductions in the manuscript.

**Revision in manuscript:**

*Page 2, Line 32-33: Estimation shows that IMO limitation of 0.1% $S_F$ in ECAs would reduce $SO_2$ emissions by 82 % by 2020 and further 23,000 tons of $SO_2$ by 2030 in European seas (Campling et al., 2013).*

4. Page 4 row 10 on hourly measurements of $PM_{2.5}$ and $PM_{10}$ by β-ray absorption should be explained. This is probably not the µg/m$^3$ measurements.

**Response:**

Thanks for your suggestion. According to the International Organization for Standardization, the β-ray absorption method is a method for the measurement of the mass of particulate matter in ambient air and is based on the absorption of beta rays by the particulate matter (ISO 10473:2000). The concentration was computed as follows:

$$C_{particles} = \frac{\Delta m \cdot S(Detection\ area, cm^2)}{q(sampling\ flow, m^3/h) \cdot t(sampling\ time, h)}$$

, where the mass per unit area of the particulate matter trapped in the filter

$$\Delta m(mg/cm^2) = \frac{\ln(N_1/N_2)}{k(absorption\ coefficient, cm^2/mg)}$$

The $N_1$ and $N_2$ represent the amount of β-ray passing through a blank filter and that trapped by particulate matter, respectively.

Similarly, several research describing the β-ray absorption method just as the equation above (Zhao et al., 2013; Zuo et al., 2017).

**Revision in manuscript:**

*Page 4, Line 20-22: Monitoring modules consist of NO, $NO_2$ and $NO_x$ measurement by an analyser, $SO_2$ detection by UV fluorometric, CO by IR absorption, $O_3$ by UV spectrophotometry, and particles by β-ray absorption (ISO 10473:2000).*

5. Suggest to change "aerosol sample" to something like e.g. "exposed filter"

**Response:**

Thanks for your suggestion. We agree that "aerosol sample" is not as equally accurate as "exposed filter". And we revise the manuscript as below.

**Revision in manuscript:**

*1) Page 5, Line 1: 2.1.4 Particle samples*

*2) Page 5, Line 3-5: The filters were exposed for 23 h (normally from 16:30 to 15:30 LST the next day, local standard time, and named after the ending date) on an 80 mm-diameter pre-fired quartz microfiber filters (CHM QF1 grade) by a Laoying Model 2030 TSP sampler.*

*3) Page 5, Line 10-12: 0.55 cm$^2$ section of each exposed filter and blank filters were measured for concentrations of organic carbon (OC) and elemental carbon (EC) by the Thermal Optical Transmission Method in a DRI 2001 organic carbon/elemental carbon (OC/EC) analyser.*

4) *Page 5, Line 19-20: 50 cm² section of each exposed filter and blank filters were extracted with 10 ml ultra-pure water in an ultrasonic bath at 4 ℃ for 30 m in.*

5) *Page 5, Line 25-26: 20 cm² section of each exposed filter and blank filters were digested with 25 ml of an 8 %-HCl/ 3 %-HNO₃ solution in an ultrasonic bath at 69 ℃ for 3 h.*

6. Explain more on why only 16 plume events are identified - the period was long and the port is described as very busy.

**Response:**

Thanks for your suggestion. The port is busy indeed, but there are several reasons for the rather few plume event. Firstly, during our measurement, there was a period when ships barely went into the port due to the New Year holidays and also due to the poor visibility from January 1 to 4, 2017. Secondly, the wind direction in JT changes quickly, and sometimes it was unfavorable for instrument to capture the ship plume. And the prevailing wind direction indicates our plumes would be mainly from the 2$^{nd}$ pool and the 3$^{rd}$ pool, of which approximately half berth were actually in construction and not in use, making our plumes quite few. Thirdly, the port is actually quite polluted (in over 50 % of days, PM$_{2.5}$ concentration was above 115 μg·m$^{-3}$), and the pollutants concentration can be rather high and may cover the existence of a ship plume event. So if the ship plume was emitted relatively far from the instrument, it would be difficult to distinguish the ship plume from background data even if the instrument captured the plume. Moreover, the measurement site is also in the vicinity of busy trucks which can be another interference. The manuscript is revised as below.

**Revision in manuscript:**

*Page 6, Line 16-31: For these time stamps, peaks in NO$_x$ along with simultaneous valleys in O$_3$ were then identified in valid data. The signals were only affirmed when there were significant peaks and clearly determinable backgrounds. Finally ship plume event were marked if the existence of ships was positive in the upwind direction of those signals. The combination of the trace gas peak time, the wind direction, and the ship traffic information (time of ships leaving and berthing) provided by marine administration in the port will enable the identification of the plume-related ship. For example, a ship plume event was identified in 5 January 2017 from 15:36 to 16:08 (Fig. 2). The timing and conditions associated with 16 positively identified ship plume event are listed in Table 1. Several situations made it more difficult to identify a ship plume event in our measurement. Firstly, there was a period when ships barely went into the port due to the New Year holidays and also due to the poor visibility from January 1 to 4, 2017. Secondly, the prevailing wind direction indicates our plumes would be mainly from the 2$^{nd}$ pool and the 3$^{rd}$ pool,*

*of which approximately half berth were actually in construction and not in use, making our plumes quite fewer than expect, let alone the fact that wind direction is actually changes quickly and sometimes unfavourable for instrument to capture the ship plume. Thirdly, the port is actually quite polluted (in over 50 % of days, $PM_{2.5}$ concentration was above 115 $\mu g \cdot m^{-3}$, see section 3.1.1), and the pollutants concentration can be rather high and may cover the existence of a ship plume event. Moreover, the measurement site is also in the vicinity of busy trucks which can be another interference.*

7. Page 6 row 16. Suggest rewrite "In addition, high concentrations of organics, metals and the compounds between are obtained in IFOs from their presence in the original crude oil." This is an unclear statement.

**Response:**

Thanks for your suggestion. We agree with you that the statement is unclear and can be quite confusing. The manuscript is revised as below.

**Revision in manuscript:**

*Page 7, Line 17-18: In addition, IFOs obtain high concentrations of organics, metals, and the compounds of organic metal from their presence in the original crude oil.*

8. Page 6 on hybrid fuels: It is important to point out that these fuels can be anything from low sulphur heavy oils to qualities close to gasoils. The important issue is that there is no standard for these fuels (e.g. ISO-standard) and the only rquirement is that the sulphur is less than specified (<0.1%)

**Response:**

Thanks for your suggestion. We agree with you that the situation of hybrid fuels should be stated clearly and the previous description is quite vague. We revise the manuscript and point out the unsupervised situation of hybrid fuels.

**Revision in manuscript:**

*Page 7, Line 31-Page 8, Line 3: Another record worth mentioning is that hybrid fuels that blend IFO and other low $S_F$ fuels to comply with $S_F$ limit are found widely used by ships operating in SECAs (Winnes et al., 2016; Zetterdahl et al., 2016), since the price of distillate fuels is an obstacle for contractors to completely abandon IFOs. However, by now ISO 8217:2017, the benchmark for the quality of marine fuels on the market, has not obtain any limits of physical and chemical parameters for hybrid fuels. It causes a large uncertainty of their qualities since there are zero formal standard for quality of hybrid fuels except the requirement of $S_F$.*

9. Page 10 row 7. The OC/EC ratio in ship emissions is probably both dependent to fuel (residual or distillate) and to engine characteristics and therefore varies a lot.

**Response:**

Thanks for your suggestion. The OC/EC ratio is indeed under influence of both fuel and engine, but several emission factor studies suggest the fact that OC/EC emission ratio is strongly distinguishable between marine combustion (typically over 10) and on-road diesel engine (typically lower than 1) (Celo et al., 2015; Khan et al., 2012; Moldanová et al., 2009; Oanh et al., 2010; Sippula et al., 2014). Therefore, the higher value of OC/EC ratio of aerosols in JT may indicate the worse influence of ship emissions than other port city like Hong Kong. We revise the manuscript with a more appropriate expression as below.

**Revision in manuscript:**

*Page 11, Line 6-10: Despite the OC/EC emission ratio dependent to both fuel type and engine, tests show that it is still strongly distinguishable between marine combustions (typically over 10) (Celo et al., 2015; Moldanová et al., 2009; Sippula et al., 2014) and on-road diesel engine (typically ranging from 0.25 to 1) (Oanh et al., 2010). In this study, the mean OC/EC ratio was 3.58, much higher than that of Thessaloniki port in Greece and Hong Kong, which indicates a worse influence of ship emissions in JT.*

10. Figure 8. There should be an explanation to what is meant by the different "classes" in the Figure caption.

**Response:**

Thanks for your suggestion. The categorization of "classes" are described in page 10 line 15 as "Samples were categorized into three batches based on the $PM_{2.5}$ limit of AQI level (HJ 633-2012) during sampling, considering the influences of ambient pollution on particulate chemical composition". And for convenience of readers, we add the explanation in the title of Figure 8.

**Revision in manuscript:**

*Figure 8: Enrichment factor of elements in $PM_{2.5}$ in JingTang Harbor. The classes are corresponding AQI level computed from $PM_{2.5}$ concentration during sampling time. The mean, minimum, and maximum concentrations of each element are also illustrated.*

**Reference**

Campling, P., Janssen, L., Vanherle, K., Cofala, J., Heyes, C., and Sander, R.: Final Report: Specific evaluation of emissions from shipping including assessment for the establishment of

possible new emission control areas in European Seas, The Flemish Institute for Technological Research NV, 74, 2013.

Celo, V., Dabek-Zlotorzynska, E., and McCurdy, M.: Chemical Characterization of Exhaust Emissions from Selected Canadian Marine Vessels: The Case of Trace Metals and Lanthanoids, Environmental Science & Technology, 49, 5220-5226, 10.1021/acs.est.5b00127, 2015.

Fu, Q., Shen, Y., and Zhang, J.: On the ship pollutant emission inventory in Shanghai port, Journal of Safety and Environment, 12, 57-64, 2012.

IMO: Third IMO GHG Study 2014, 2015.

ISO 10473:2000, Ambient air — Measurement of the mass of particulate matter on a filter medium — Beta-ray absorption method.

ISO 8217:2017. Petroleum products-Fuels (class F)-Specifications of marine fuels.

Jalkanen, J. P., Johansson, L., and Kukkonen, J.: A comprehensive inventory of the ship traffic exhaust emissions in the Baltic Sea from 2006 to 2009, Ambio, 43, 311-324, 10.1007/s13280-013-0389-3, 2014.

Jalkanen, J. P., Johansson, L., and Kukkonen, J.: A comprehensive inventory of ship traffic exhaust emissions in the European sea areas in 2011, Atmospheric Chemistry and Physics, 16, 71-84, 10.5194/acp-16-71-2016, 2016.

Khan, M. Y., Giordano, M., Gutierrez, J., Welch, W. A., Asa-Awuku, A., Miller, J. W., & Cocker, D. R. (2012). Benefits of Two Mitigation Strategies for Container Vessels: Cleaner Engines and Cleaner Fuels. ENVIRONMENTAL SCIENCE & TECHNOLOGY, 46(9), 5049-5056. doi:10.1021/es2043646

Kattner, L., Mathieu-Uffing, B., Burrows, J. P., Richter, A., Schmolke, S., Seyler, A., & Wittrock, F. (2015). Monitoring compliance with sulfur content regulations of shipping fuel by in situ measurements of ship emissions. Atmospheric Chemistry and Physics, 15(17), 10087-10092. doi:10.5194/acp-15-10087-2015

Li, C., Yuan, Z., Ou, J., Fan, X., Ye, S., Xiao, T., Shi, Y., Huang, Z., Ng, S. K. W., Zhong, Z., and Zheng, J.: An AIS-based high-resolution ship emission inventory and its uncertainty in Pearl River Delta region, China, Science of the Total Environment, 573, 1-10, 10.1016/j.scitotenv.2016.07.219, 2016.

Loov, J. M. B., Alfoldy, B., Gast, L. F. L., Hjorth, J., Lagler, F., Mellqvist, J., . . . Borowiak, A. (2014). Field test of available methods to measure remotely SOx and NOx emissions from ships. Atmospheric Measurement Techniques, 7(8), 2597-2613. doi:10.5194/amt-7-2597-2014

Lv, Z., Liu, H., Ying, Q., Fu, M., Meng, Z., Wang, Y., Wei, W., Gong, H., and He, K.: Impacts of shipping emissions on PM2.5 air pollution in China, Atmospheric Chemistry and Physics

Discussion, https://doi.org/10.5194/acp-2018-540, 2018.

Moldanová, J., Fridell, E., Popovicheva, O., Demirdjian, B., Tishkova, V., Faccinetto, A., and Focsa, C.: Characterisation of particulate matter and gaseous emissions from a large ship diesel engine, Atmospheric Environment, 43, 2632-2641, 10.1016/j.atmosenv.2009.02.008, 2009.

5  Oanh, N. T. K., Thiansathit, W., Bond, T. C., Subramanian, R., Winijkul, E., and Paw-armart, I.: Compositional characterization of PM2.5 emitted from in-use diesel vehicles, Atmospheric Environment, 44, 15-22, 10.1016/j.atmosenv.2009.10.005, 2010

Sippula, O., Stengel, B., Sklorz, M., Streibel, T., Rabe, R., Orasche, J., Lintelmann, J., Michalke, B., Abbaszade, G., Radischat, C., Groeger, T., Schnelle-Kreis, J., Harndorf, H., and Zimmermann,

10  R.: Particle Emissions from a Marine Engine: Chemical Composition and Aromatic Emission Profiles under Various Operating Conditions, Environmental Science & Technology, 48, 11721-11729, 10.1021/es502484z, 2014.

Winnes, H., Moldanova, J., Anderson, M., and Fridell, E.: On-board measurements of particle emissions from marine engines using fuels with different sulphur content, Proceedings of the

15  Institution of Mechanical Engineers Part M-Journal of Engineering for the Maritime Environment, 230, 45-54, 10.1177/1475090214530877, 2016.

Yang, M., Bell, T. G., Hopkins, F. E., & Smyth, T. J. (2016). Attribution of atmospheric sulfur dioxide over the English Channel to dimethyl sulfide and changing ship emissions. Atmospheric Chemistry and Physics, 16(8), 4771-4783. doi:10.5194/acp-16-4771-2016

20  Zetterdahl, M., Moldanova, J., Pei, X., Pathak, R. K., and Demirdjian, B.: Impact of the 0.1% fuel sulfur content limit in SECA on particle and gaseous emissions from marine vessels, Atmospheric Environment, 145, 338-345, 10.1016/j.atmosenv.2016.09.022, 2016.

Zhao, X., Pan, J. X., Liu, B., and Chen, P.: Research of measuring technology for pm2.5 based on the β-ray absorption method. Application of Electronic Technique, 2013. (in Chinese)

25  Zuo, J. and Zhao Z.: Research of PM2.5 Monitoring Key Technology Based on β-Ray Method. Instrument Technique and Sensor, 2017. (in Chinese)

---

## Editor Comment (EC1) · Markus Quante (Editor) · 25 Mar 2019

Dear authors,

thanks for the revised manuscript, which is now accepted. The answers to the reviewer comments and respective changes in the manuscript are convincing. Thanks also for the supporting additional references. Little language issues (definite article, plural) will be taken care of by the Copernicus "copy-editing lite" process. I thank the authors and the reviewers for a clear and efficient discussion.

Regards, Markus